# Transcriptional landscape of myogenesis from human pluripotent stem cells reveals a key role of TWIST1 in maintenance of skeletal muscle progenitors

In Young Choi[1,2†], Hotae Lim[1,3†], Hyeon Jin Cho[4‡], Yohan Oh[1§], Bin-Kuan Chou[1,5], Hao Bai[1,5], Linzhao Cheng[5], Yong Jun Kim[6], SangHwan Hyun[1,3], Hyesoo Kim[1,7], Joo Heon Shin[4], Gabsang Lee[1,7,8*]

[1]The Institute for Cell Engineering, Johns Hopkins University, School of Medicine, Baltimore, United States; [2]Department of Medicine, Graduate School, Kyung Hee University, Seoul, Republic of Korea; [3]College of Veterinary Medicine, Chungbuk National University, Chungbuk, Republic of Korea; [4]Lieber Institute for Brain Development, Johns Hopkins Medical Campus, Baltimore, United States; [5]Division of Hematology, Department of Medicine, Johns Hopkins University, School of Medicine, Baltimore, United States; [6]Department of Pathololgy, College of Medicine, Kyung Hee University, Seoul, Republic of Korea; [7]Department of Neurology, Johns Hopkins University, School of Medicine, Baltimore, United States; [8]The Solomon H. Synder Department of Neuroscience, Johns Hopkins University, School of Medicine, Baltimore, United States

*For correspondence:
glee48@jhmi.edu

†These authors contributed equally to this work

Present address: ‡University of Maryland, College Park, United States; §Department of Medicine, College of Medicine, Hanyang University, Seoul, Republic of Korea

Competing interests: The authors declare that no competing interests exist.

**Abstract** Generation of skeletal muscle cells with human pluripotent stem cells (hPSCs) opens new avenues for deciphering essential, but poorly understood aspects of transcriptional regulation in human myogenic specification. In this study, we characterized the transcriptional landscape of distinct human myogenic stages, including OCT4::EGFP+ pluripotent stem cells, MSGN1::EGFP+ presomite cells, PAX7::EGFP+ skeletal muscle progenitor cells, MYOG::EGFP+ myoblasts, and multinucleated myotubes. We defined signature gene expression profiles from each isolated cell population with unbiased clustering analysis, which provided unique insights into the transcriptional dynamics of human myogenesis from undifferentiated hPSCs to fully differentiated myotubes. Using a knock-out strategy, we identified TWIST1 as a critical factor in maintenance of human PAX7::EGFP+ putative skeletal muscle progenitor cells. Our data revealed a new role of TWIST1 in human skeletal muscle progenitors, and we have established a foundation to identify transcriptional regulations of human myogenic ontogeny (online database can be accessed in http://www.myogenesis.net/).

## Introduction

Stem cell biology using human pluripotent stem cells (hPSCs), including embryonic stem cells (hESCs) and induced pluripotent stem cells (hiPSCs), is providing unprecedented opportunities for the study of human development by recapitulating embryogenesis (*Tchieu et al., 2017*; *Irie et al., 2015*; *van de Leemput et al., 2014*; *Qi et al., 2017*; *Guo et al., 2017*). Directed specification of hESCs/hiPSCs is especially advantageous for certain cell types because it produces large quantities of otherwise extremely rare cell populations during development (e.g. skeletal muscle stem/progenitor cells). Often such cellular subtypes are critical for the formation of a tissue type, and they can be

important for a better understanding of cellular ontogeny from embryonic stage to postnatal stage. In particular, compartmentalization of the serial developmental processes using hPSCs in vitro can deliver unprecedented opportunity to gain insights into the transcriptional continuum of a specific cellular lineage. For example, the analysis of global transcriptional expression in myogenic development in a 'step-wise' platform can provide invaluable information regarding the patterns of transcriptional dynamics related to human skeletal muscle specification processes. Furthermore, while there are many rare genetic disorders with musculoskeletal symptoms, we have limited tools to gain a better understanding of the disease mechanisms.

During embryonic myogenesis, skeletal muscle is generated through multiple cellular stages, including pluripotent preimplanted embryos, somite cells, skeletal muscle stem/precursor cells, proliferating myoblasts, and multinucleated myotubes. While previous studies reported the generation of in vitro skeletal muscle cells with overexpression of exogenous myogenic transcription factors (TFs) into fibroblasts and stem cells (*Darabi et al., 2012*; *Tedesco et al., 2012*; *Albini et al., 2013*; *Salani et al., 2012*; *Davis et al., 1987*; *Magli and Perlingeiro, 2017*), they did not provide a step-wise transcriptional blueprint of myogenic specification. Recently, several reports were published that effectively directed hPSCs to human skeletal muscle cell fates (*Chal et al., 2015*; *Choi et al., 2016*; *Shelton et al., 2014*), which can be a reasonable platform to tease out the transcriptional trajectory of human myogenesis processes. However, the heterogenous cellular populations in the myogenic cell culture hamper further cellular isolation of specific myogenic cells for detailed studies.

Identification of the molecular regulators for the generation of human skeletal muscle cells is important for discovering molecular mechanisms of human myogenesis. While OCT4 is one of the POU transcription factors that is essential for maintaining pluripotency and self-renewal of PSCs, MSGN1 has a key function in generation of paraxial mesoderm cells (*Loh et al., 2006*; *Fior et al., 2012*). MYOG is a skeletal muscle-specific transcription factor in the basic helix loop helix class of DNA-binding proteins and has a critical role in skeletal muscle differentiation during embryogenesis (*Nabeshima et al., 1993*; *Bentzinger et al., 2012*; *Hasty et al., 1993*; *Brunetti and Goldfine, 1990*; *Magli and Perlingeiro, 2017*). Another major gene is PAX7, a member of the paired box (PAX) family of transcription factors, which has important roles in skeletal muscle development, maintenance, and regeneration (*Bentzinger et al., 2012*; *Seale et al., 2000*; *Kassar-Duchossoy et al., 2005*; *Parker et al., 2003*; *Biressi et al., 2007*; *Buckingham and Relaix, 2007*). Previous reports with genetic reporter mice showed the transcriptional profiles of PAX3 or PAX7 expressing cells in embryonic and postnatal muscle tissues (*Relaix et al., 2005*; *Tichy et al., 2018*), which presented developmental transcriptional changes in a specific population.

Recently, the CRISPR/Cas9 system has been developed and used widely in genetic modification of hPSCs due to its simplified design and feasibility (*Mali et al., 2013*; *Cong et al., 2013*). For instance, a 'knock-in' strategy to generate a genetic reporter hPSC line using the CRISPR/Cas9 system is now feasible and enables us to purify a homogenous cell population by prospective isolation (*Ganat et al., 2012*; *Tchieu et al., 2017*; *Mukherjee et al., 2018*). Moreover, multiple genetic reporter hPSC lines for stage-specific transcription factors can provide a step-wise transcriptional landscape for a certain lineage.

In this study, to systematically investigate the transcriptional blueprint for developing human skeletal muscle cells and to gain comprehensive insights into the molecular signatures of putative skeletal muscle stem/progenitors, we conducted step-wise isolations of stage-specific cellular subtypes during muscle differentiation in vitro and performed global gene expression analysis. Using our human genetic reporter PSC lines and a newly devised method for myotube enrichment, we isolated five distinct cell types in human embryonic myogenesis, including OCT4::EGFP+ embryonic stem cells, MSGN1::EGFP+ presomite cells, PAX7::EGFP+ putative skeletal muscle stem/precursor cells, MYOG::EGFP+ myoblast cells, and multinucleated myotubes (*Loh et al., 2006*; *Fior et al., 2012*; *Nichols et al., 1998*; *Seale et al., 2000*; *Hasty et al., 1993*). The clustering analysis presented transcriptional changes during in vitro muscle generation, and we identified several transcription factors that have key roles in human myogenesis in vitro. One of the transcription factors we identified is TWIST1, which has not been extensively studied in human muscle biology and relevant genetic diseases.

## Results

### Generation of stage-specific genetic reporter hPSC lines to simulate human embryonic myogenesis in vitro

During human embryonic myogenesis, several key marker genes are known to play significant roles in each stage (*Figure 1A*), for example, *OCT4* and *NANOG* in pluripotent stem cells, *MSGN1* and *TBX6* in presomite cells (*Chapman and Papaioannou, 1998*; *Fior et al., 2012*; *Loh et al., 2006*; *Thomson et al., 1998*), *PAX7* in putative myogenic stem/progenitor cells, and *MYOD1* and *MYOG* in myoblasts before myotube formation (*Nabeshima et al., 1993*; *Seale et al., 2000*; *Hasty et al., 1993*; *Kassar-Duchossoy et al., 2005*). Previously, we have developed an in vitro myogenic specification protocol directing hPSCs into human skeletal muscle cells through the GSK3β and Notch signal inhibition pathway (*Choi et al., 2016*). We used this protocol to test whether differentiating hPSC cells express stage-specific myogenic transcription factors. Time course expression of each gene mentioned above was profiled using quantitative Real-Time PCR (qRT-PCR) analysis for the first 30 days of differentiation (*Figure 1—figure supplement 1A*). Expression levels of pluripotency markers, *OCT4* and *NANOG,* were high in undifferentiated hESCs, but decreased rapidly upon initiation of muscle specification. Within 4 days of myogenic specification, the expression of mesoderm markers *T (Brachyury)*, *MIXL1*, and presomite markers *MSGN1* and *TBX6* was induced, while the expression levels of *PAX7*, *MYF5, MYOD1* and *MYOG* gradually increased around day 20. For the characterization between MSGN1 and PAX7, we performed the gene expression profiles of *PAX3*, *MEOX1*, *FOXC1*, *FOXC2*, *PARAXIS*, and *SCLERAXIS* during in vitro myogenesis. *PAX3*, *MEOX1*, *FOXC1*, and *FOXC2* gene started their gene expression at Day 4, and had a peak between Day 6 and Day 8 which imply that intermediate somite stage fills the gap between MSGN1+ stage and PAX7+ stage. To determine protein expression levels, we performed immunostaining in each stage with OCT4, TBX6, PAX7, MYOG, MYHs (MF20), and ACTN1 (α-actinin) antibodies (*Figure 1B*). Distinct protein expression patterns were observed during our in vitro myogenic specification: OCT4 expressing cells were 96.42 ± 2.55% of undifferentiated hESCs (mean ± SEM); at day 4, 87.78 ± 4.46% of the cell population expressed TBX6; at day 20, 31.72 ± 5.78% of the cell population expressed PAX7; at day 25, 53.30 ± 6.39% of the cell population expressed MYOG; at day 40, 87.99 ± 3.64% of the cell population expressed MF20. Multinucleated and striated myofibers were generated with expression of the myofiber marker, α-actinin. Notably, cardiac troponin T (cTnT) and smooth muscle alpha actin (SMAA)-positive cells were hardly detected (data not shown), demonstrating that there is almost no contamination of cardiac muscle or smooth muscle lineage. Taken together, these data demonstrated that using our skeletal muscle protocol, hPSCs can be directed to skeletal muscle lineages with the expression of key marker genes.

To trace and isolate a homogenous cell population during in vitro human embryonic myogenesis, we generated two hESC genetic reporter lines for PAX7 and MYOG with the 2A-EGFP knock-in system (*Figure 1—figure supplement 1B–C*). PAX7 and MYOG are well known marker genes for skeletal muscle stem/progenitor cells and myoblasts, respectively, and both are expressed in our in vitro myogenic specification process. The genetic reporter lines were established using the CRISPR/Cas9 system (*Cong et al., 2013*; *Mali et al., 2013*) as described previously by our group (OCT4::EGFP and MSGN1::EGFP lines) (*Choi et al., 2016*; *Kim et al., 2014*). Each reporter clone was validated by FACS analysis (*Figure 1C*), and clones with detectable levels of EGFP expression were chosen for further studies. In each stage, the OCT4::EGFP reporter line showed EGFP percentages of 80.97 ± 2.09% at day 0; MSGN1::EGFP, 94.98 ± 0.60% at day 4; PAX7::EGFP, 17.80 ± 1.44% at day 33; MYOG::EGFP, 11.81 ± 1.41% at day 35 (mean ± SEM). Reporting ability was verified with FACS-sorted cells by analyzing the enrichment levels of the related marker gene expression with qRT-PCR (*Figure 1—figure supplement 1D–G*). PAX7::EGFP+ cells exhibited enrichments of *MYF5* (24.04 ± 2.06, mean ± SEM of fold changes), *MYOD1* (23.37 ± 1.77), and *MYOG* (22.98 ± 2.94) expression compared to PAX7::EGFP- cells, while MYOG::EGFP+ cells also showed increased levels of *MYOD1*, *MEF2C* and *MYH2* expression (149.6 ± 36.77, 69.83 ± 16.18 and 1566 ± 808.5 fold changes, respectively) compared to MYOG::EGFP- cells. To confirm the cellular identity of the PAX7::EGFP+ cells generated by our skeletal muscle differentiation protocol, we performed qRT-PCR with primer sets specific for neural and neural crest lineages such as *SOX10*, *PAX6*, and antibody staining of SOX10 and AP2 (*Figure 1—figure supplement 2A–B*). Both PAX7::EGFP+ cells and PAX7::EGFP- cells showed significantly low levels of gene expression compared to hESC-derived

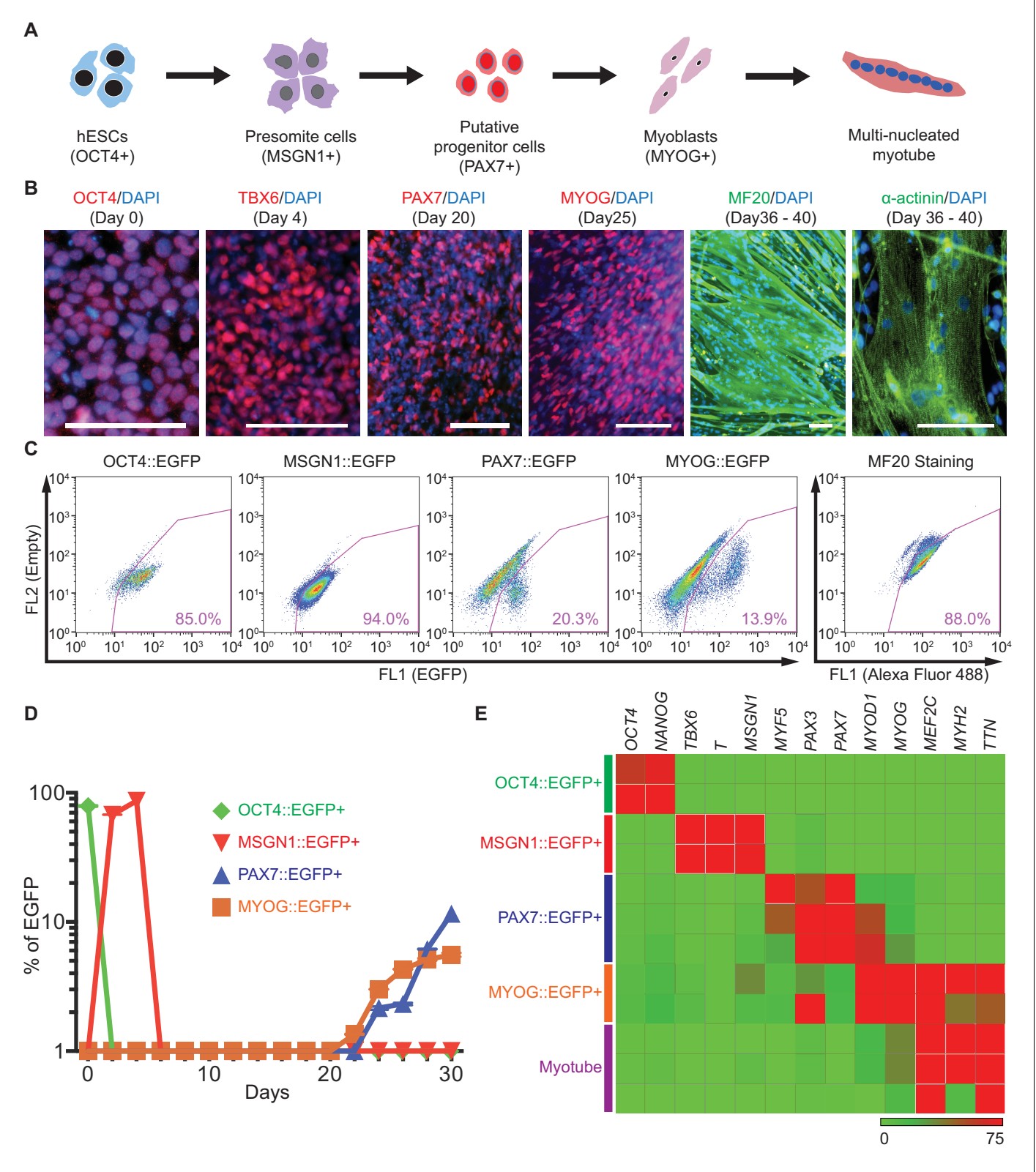

**Figure 1.** Generation and characterization of genetic reporter hPSC lines for stage-specific markers during human skeletal muscle specification. (**A**) Schematic illustration of the embryonic myogenesis of hPSCs with stage-specific marker genes. (**B**) Immunocytochemistry of OCT4, TBX6, PAX7, MYOG, MF20 and α-actinin during in vitro muscle differentiation. (bars, 100 μm) (**C**) FACS plots of multiple reporter lines during in vitro muscle differentiation with two chemical compounds and expression of MF20, myotube marker. (**D**) Plot of EGFP percentage for each marker in multiple reporter lines during

*Figure 1 continued on next page*

*Figure 1 continued*

in vitro muscle differentiation. Values were detected every 2 days. (E) Heatmap of stage-specific marker genes expression in FACS-sorted positive population for each marker in multiple reporter lines (OCT4::EGFP+ cells at day 0, MSGN1::EGFP+ cells at day 4, PAX7::EGFP+ cells between day 26 and day 30, MYOG::EGFP+ cells between day 34 and day 37, and myotube isolation at day 36–40. The maximum values in each column were adjusted to 100.).

The online version of this article includes the following figure supplement(s) for figure 1:

**Figure supplement 1.** Generation and validation of stage-specific hPSC reporter lines for myogenesis.

**Figure supplement 2.** Characterization of PAX7::EGFP+ cells.

neural crest cells (as a positive control). In the protein level, we did not detect any SOX10+ cells, or AP2+ cells in the PAX7::EGFP+ cells via antibody staining (*Figure 1—figure supplement 2C*). Furthermore, we confirmed the most of the PAX7::EGFP+ cells express PAX7 protein, but not MYOD1 and MYOG proteins (*Figure 1—figure supplement 2D*), while 33.17 ± 3.26% of Ki67+ cells are found in PAX7 + cells and 4.92 ± 0.79% of Ki67+ cells in MYOG + cells (*Figure 1—figure supplement 2E*). Single-cell transcription analysis data showed that some of PAX7::EGFP+ cells already show *MYOG* expression as well as other marker genes, including *MYF5* and *MYOD1*, while most of the PAX7::EGFP+ cells have high level of *PAX7* expression (*Figure 1—figure supplement 2F*). For the confirmation of fusion abilities with the PAX7::EGFP+ cells, post-sorted cells were induced to the terminal differentiation or myotube formation, and showed great fusion ability with spontaneous twitching (*Video 1*). FACS analysis followed by intracellular staining with MF20 antibody showed 85.28 ± 2.04% of expression at day 23 (*Figure 1C*).

Next, we examined the time-course expression levels of *OCT4, MSGN1, PAX7* and *MYOG* in each genetic reporter line (*Figure 1D*). The OCT4::EGFP line had a peak EGFP percentage at day 0, MSGN1::EGFP line peaked at days 2 and 4, and both PAX7::EGFP and MYOG::EGFP lines showed EGFP+ cells at day 20, which gradually increased and reached highest levels at day 30. These results were corroborated by the gene expression profiles measured by qRT-PCR (*Figure 1—figure supplement 1A*) and demonstrated the reliable reporting ability of our genetic reporter hPSC lines.

During skeletal muscle specification with each reporter cell line, EGFP+ cells were harvested using FACS purification at their highest peak of expression (OCT4::EGFP+ cells at day 0, MSGN1::EGFP+ cells at day 4, PAX7::EGFP+ cells between day 26 and day 30, and MYOG::EGFP+ cells between day 34 and day 37) and subjected to gene expression profiling. For multinucleated myotube isolation from the heterogeneous culture, we devised a new method based on differential detachment rates between myotubes and non-myotube cells upon trypsin treatment. Non-myotube cells tend to detach faster than myotubes, which enabled us to remove non-myotube cells (mostly myoblast-like cells) first, followed by harvesting remnant multinucleated myotubes a few minutes later (*Figure 1—figure supplement 1H–I*). We confirmed that the remnant cell population showed enriched gene expression levels of *DMD* (32.69 ± 1.28 fold changes), *TTN* (27.42 ± 1.26), and *MYH2* (21.40 ± 0.95) compared to the detached cell population (*Figure 1—figure supplement 1J*). Each FACS purified population was validated using qRT-PCR (*Figure 1E*), and the results confirmed that each cell population showed high levels of expression of the representative marker genes for the corresponding stage of differentiation: *OCT4* and *NANOG* in OCT4::EGFP+ cells, *TBX6*, *T*, and *MSGN1* in MSGN1::EGFP+ cells, *PAX3* and *PAX7* in PAX7::EGFP+ cells, *MYOD1, MYOG, MEF2C, MYH2,* and *TTN* in MYOG::EGFP+ cells, and *MEF2C, MYH2,* and *TTN* in isolated myotubes.

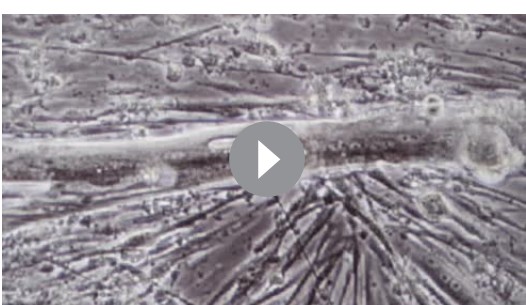

**Video 1.** Twitching multinucleated myotubes derived from FACS-purified PAX7::EGFP+ cells.
https://elifesciences.org/articles/46981#video1

## Genome-wide transcriptome analysis of FACS-purified cell populations during human myogenesis

In order to comprehensively examine stage-specific gene expression profiles during in vitro

human myogenesis at the genome-wide level, we carried out RNA Sequencing (RNA-seq) of the purified cells from five different stages of myogenesis: OCT4::EGFP+ cells, MSGN1::EGFP+ cells, PAX7::EGFP+ cells, MYOG::EGFP+ cells, and Myotubes. Using the isolation strategy outlined earlier, we prepared four populations with genetic reporter lines and one with myotube separation strategy (four independent biological replicates). On average, 83 million total reads and 34 million aligned reads were produced per sample, and sequencing reads were evenly distributed throughout the whole span of transcripts. Universal genes, such as *GAPDH* and *ACTB*, were evenly expressed across samples, whereas stemness markers (*OCT4*, *NANOG*, and *SOX2*), or pan somite markers (*MSGN1*, *T* (*Brachyury*), and *TBX6*), and myogenesis markers (*PAX7*, *PAX3*, *MYF5*, *MYOD1*, *MYOG*, *TTN*, *MYH3*, *MYH2*, and *MYH7*) were expressed only in subsets of groups (*Figure 2A*, *Figure 2—figure supplement 1A*, and *Supplementary file 1*). OCT4::EGFP+ cells expressed high levels of *OCT4*, *NANOG*, and *SOX2*, and MSGN1::EGFP+ cells were enriched with genes for the presomite stage, such as *MSGN1*, *T*, and *TBX6*. PAX7::EGFP+ cells had high expression of *PAX7*, *PAX3*, and *MYF5*, and MYOG::EGFP+ cells had enriched gene expression of *MYOG*, *MYOD1*, *TTN* and *MYH3*. While *TTN* and *MYH3* were expressed in both MYOG::EGFP+ cells and Myotubes, *MYH2* (fast twitching fiber marker gene) (*Figure 2—figure supplement 1A*) was expressed only in Myotube samples, which represent secondary myogenesis (*Parker et al., 2003*; *Biressi et al., 2007*).

To evaluate the whole-transcriptome data set of individual groups, we compared differentially expressed genes across five stage-specific populations and grouped them based on their gene expression patterns. Unsupervised hierarchical clustering analysis of global gene expression showed correlations of five groups with clustering (*Figure 2B*). The heatmap showed 5361 differentially expressed genes through the groups with unsupervised hierarchical clustering (*Figure 2—figure supplement 1B*). The PAX7::EGFP+ population clustered with the MYOG::EGFP+ population and Myotubes, whereas the OCT4::EGFP+ population and MSGN1::EGFP+ cells clustered together in another branch away from the former group. To obtain an overview of the transcriptional association, principal component analysis (PCA) was performed, and the genetic distance and relatedness between the five cell populations were illustrated (*Figure 2C*). This analysis showed the transcriptional direction from pluripotent stem cells toward somite like cells, myogenic cells, and myotubes in three axes, named 'virtual myotime'. In order to better understand the important biological processes during human skeletal muscle development, we performed statistical Gene Ontology (GO) analysis (*Figure 2—figure supplement 1C*). The GO terms of Anterior/posterior pattern specification (GO:0009952, p value, 2.02E-10) in MSGN1::EGFP+ cells and striated muscle tissue development (GO:0014706, 2.85E-08) in Myotubes showed statistical significance and revealed the transcriptional directionality from pluripotent stem cells into multinucleated myotubes. We next examined differentially expressed TFs between groups for the enrichment of GO biological process (*Figure 2D*). The GO analysis showed enrichment of Embryo Development (GO:0009790, 1.37E-06), Stem Cell Population Maintenance (GO:0019827, 2.44E-06), and Maintenance of Cell Number (GO:0098727, 2.77E-06) in OCT4::EGFP+ cells; Embryo Organ Development (GO:0048568, 6.11E-12), Regulation of Cell Differentiation (GO:0045595, 3.57E-11), and Anterior/posterior Pattern Specification (GO:0009952, 2.02E-10) in MSGN1::EGFP+ cells; and Muscle Structure Development (GO:0061061, 2.35E-11), Skeletal Muscle Tissue Development (GO:0007519, 1.76E-08), and Striated Muscle Tissue Development (GO:0014706, 2.85E-08) in Myotubes. The enriched GO terms in MYOG::EGFP+ cells included Regulation of Biological Process (GO:0050789, 1.04E-09), Skeletal Muscle Tissue Development (GO:0007519, 1.08E-09), and Skeletal Muscle Organ Development (GO:0060538, 1.62E-09), while PAX7::EGFP+ cells involved GO terms related to muscle development, such as Muscle Structure Development (GO:0061061, 3.42E-12), as well as general differentiation GO terms, including Regulation of Cell Differentiation (GO:0045595, 3.92E-14) and Regulation of Cell Proliferation (GO:0042127, 4.45E-12).

To further investigate the relationship between PAX7 and MYOG, we compared expression levels of a set of TFs between PAX7::EGFP+ cells and MYOG::EGFP+ cells (*Figure 2—figure supplement 1D* and *Supplementary file 2*). Many upregulated TFs in PAX7::EGFP+ cells were related to signaling pathways such as *FOS*, *ID3*, *ID1*, *JUN*, *PAX7*, *JUNB*, and *FOSB*, while TFs in MYOG::EGFP+ cells included myoblast-specific markers such as *MYOG*, *MEF2C*, *MYOD1*, *MYF6*, and *FOXO4*. To investigate the similarities and dissimilarities in the global gene expression between PAX7::EGFP+ cells and MYOG::EGFP+ cells, a Venn diagram was plotted (*Figure 2E* and *Supplementary file 2*). Out of 3170 genes highly upregulated over OCT4::EGFP+ cells, PAX7::EGFP+ cells and MYOG::EGFP+

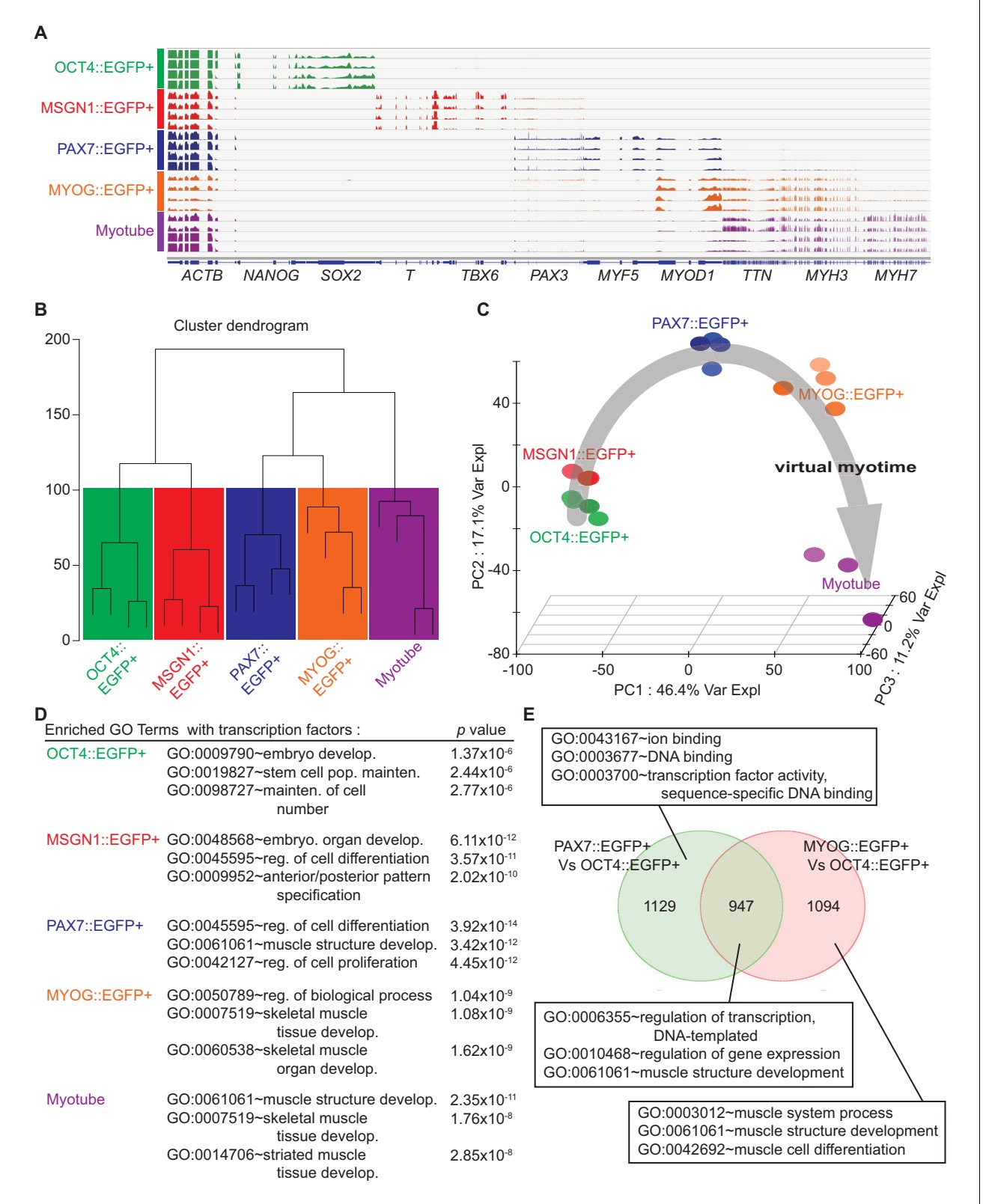

**Figure 2.** Cell population network analysis with RNA-sequencing data from five different cell types. Each sample of cell population was collected as described in *Figure 1E*. (A) Coverage profiles of total RNA from each group: *NANOG, SOX2, T, TBX6, PAX3, MYF5, MYOD1, TTN, MYH3, MYH7*, and universally expressed gene *ACTB*. (B) Dendrogram of RNA-seq result produced by hierarchical clustering of five cell populations. (C) Principal component analysis (PCA) plot of five cell populations during in vitro myogenesis. Gray arrow indicates 'Virtual myotime' from OCT4::EGFP+ to

*Figure 2 continued*

Myotube samples. (D) Enriched Gene ontology (GO) terms with transcription factors that had high p values in five cell populations from RNA-seq data. (E) Venn diagram of the upregulated genes and their GO terms in PAX7::EGFP+ cells and MYOG::EGFP+ cells compared to OCT4::EGFP+ cells.

The online version of this article includes the following figure supplement(s) for figure 2:

**Figure supplement 1.** Transcriptome in cell populations at each stage of embryonic myogenesis.
**Figure supplement 2.** Differential gene expression analysis among PAX7::EGFP+ cells, MOYG::EGFP+ cells, and Myotubes.

cells had 947 shared genes with GO terms of DNA-templated Regulation of Transcription (GO:0006355, 3.69E-15), Regulation of Gene Expression (GO:0010468, 1.63E-14), and Muscle Structure Development (GO:0061061, 1.09E-11). Meanwhile, 1129 genes upregulated in PAX7::EGFP+ cells were classified to GO terms such as Ion Binding (GO:00431677, 56E-06), DNA Binding (GO:0003677, 7.14E-04), and Sequence-specific DNA Binding Transcription Factor Activity (GO:0003700, 1.42E-03). GO analysis for 1094 genes highly expressed in MYOG::EGFP+ cells revealed significant enrichment of genes related to Muscle System Process (GO:0003012, 4.84E-28), Muscle Structure Development (GO:0061061, 3.03E-26), and Muscle Cell Differentiation (GO:0042692, 7.34E-19). Furthermore, we classified highly enriched genes specifically in PAX7:: EGFP+ cells, MYOG::EGFP+ cells, and Myotubes. For the molecular arrangement, we compared the FPKMs of all genes that were differentially expressed across all samples. Venn diagram showed clear separation between PAX7::EGFP+ cells and Myotubes, while there are overlapped genes in PAX7:: EGFP+ cells vs. MYOG::EGFP+ cells, and MYOG::EGFP+ cells vs. Myotubes (*Figure 2—figure supplement 2A*). Out of 2336 genes upregulated over all samples, 420 genes have specific expression patterns only in PAX7::EGFP+ cells, which were classified into GO terms involved in some signaling pathways including 'PI3K-Akt signaling pathway', 'Wnt signaling', and 'Wnt signaling and pluripotency' (*Figure 2—figure supplement 2B*).

These data were confirmed via a phosphorylation antibody blot (R and D, ARY003B), presenting that significantly increased levels of phosphorylation in the CREB and β-catenin in the PAX7::EGFP+ cells over the MYOG::EGFP+ cells (*Figure 2—figure supplement 2C*). These data reflected that PAX7::EGFP+ cells have a distinctively different transcriptional nature from MYOG::EGFP+ cells.

## Clustering analysis showed unique transcriptional expression profiles

To uncover distinct transcriptional changes during in vitro human skeletal muscle specification, we employed K-mean clustering in order to group genes with significant changes in time-dependent expressions into 10 clusters with median values for a total of 22,939 gene expressions (*Figure 3— figure supplement 1A* and *Supplementary file 3*). To validate these categories, we visualized transcriptional levels of a top 50 gene list in each cluster, and their expression profiles were used to help establish milestones along the differentiation time course. To validate the clustering analysis, we chose well-known genes for each cluster and examined their expression levels in five stage-specific cell populations during myogenesis (*Figure 3A* and *Supplementary file 3*). All selected marker genes were highly enriched in their corresponding cell type groups, supporting the accuracy of K-mean clustering. The five clusters were chosen as these are representative to the known stages of myogenesis, and hoping to uncover previously unknown transcriptional regulator(s) in each stage. Several genes have spike during the skeletal muscle generation were selected for the further investigation (*Figure 3B*). Each cluster presented unique distinctive transcriptional changes during in vitro myogenesis. For example, Cluster 1 was comprised of genes upregulated in MSGN1::EGFP+ cells, which include distinct presomite marker genes such as *TBX6*, *MSGN1*, *T*, and *MESP1*, without overlapping expression of marker genes of pluripotent stem cell markers or myoblast markers. Also, expression of *CDX1/2* and *GSC* genes in Cluster one were enriched exclusively in MSGN1::EGFP+ cells (*Ikeya and Takada, 2001*; *Niehrs et al., 1994*). While Cluster 2 included genes related to myotubes such as *MYH1*, *MYL2*, *MYH2*, *ITGB1*, and *TMEM182* (*Leikina et al., 2018*; *Mascarello et al., 2016*; *Stuart et al., 2016*; *Schwander et al., 2003*), Cluster 3 had known myoblast marker genes with upregulated expression patterns in both PAX7::EGFP+ cells and MYOG::EGFP+ cells, such as *SIX4*, *PITX3*, *MYF6*, and *MSTN* (*Bentzinger et al., 2012*; *Chang and Kioussi, 2018*; *McFarlane et al., 2011*). Cluster 4 showed highly upregulated gene expression profiles in PAX7:: EGFP+ cells, indicating the presence of early myoblast marker genes, such as *PAX7* and *MYF5*

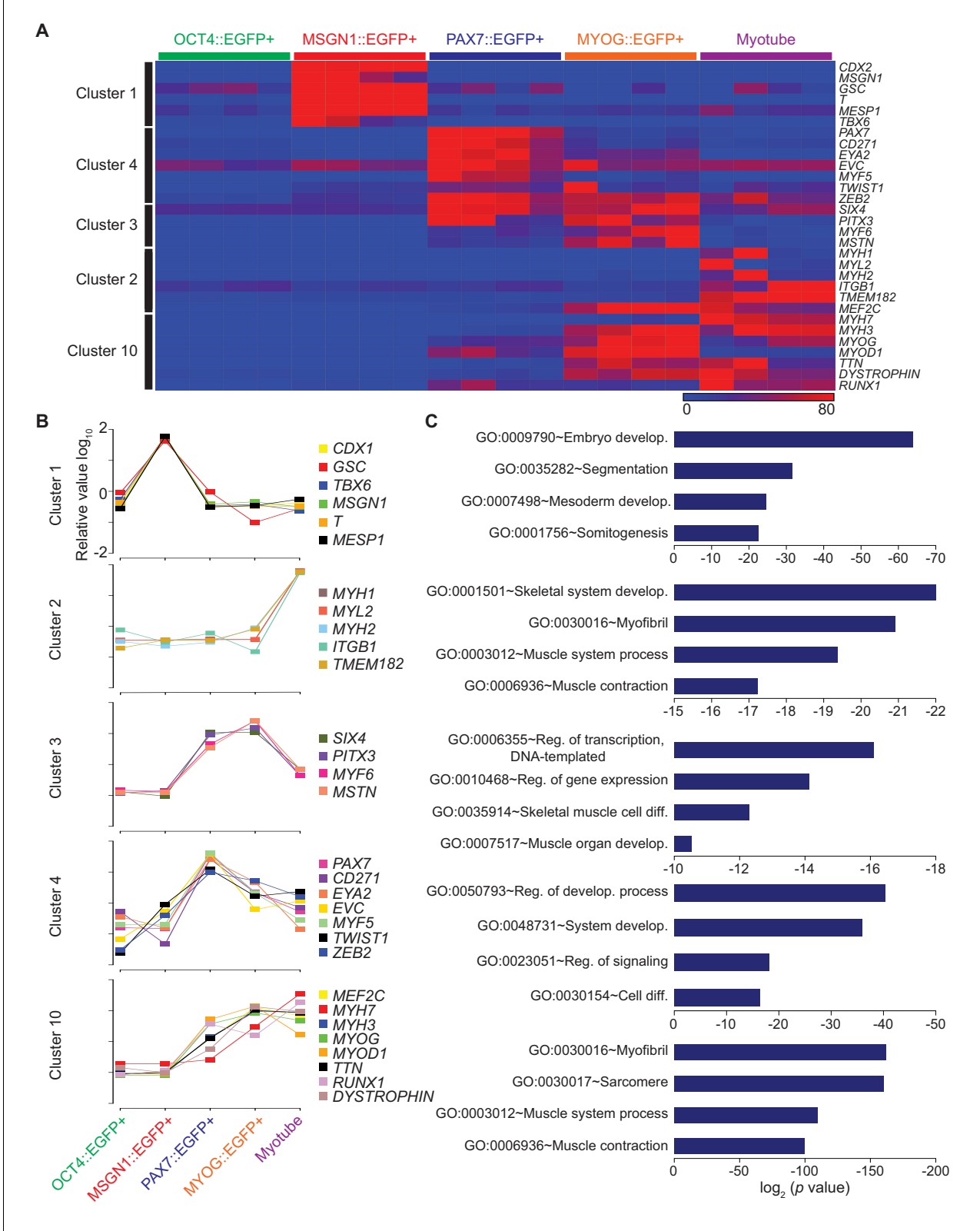

**Figure 3.** Clustering analysis of differential gene expression patterns. (**A**) Validation of clustering analysis from RNA-seq data. The heatmap of gene sets from clusters in each cell population during myogenesis showed similar expression patterns of genes in the same cluster. (**B**) The expression patterns of multiple genes from each cluster; Cluster 1, Cluster 2, Cluster 3, Cluster 4, and Cluster 10 during muscle differentiation and their GO terms (**C**).
*Figure 3 continued on next page*

*Figure 3 continued*

The online version of this article includes the following figure supplement(s) for figure 3:

**Figure supplement 1.** Ten clustering patterns from RNA-seq data analysis.

(*Gianakopoulos et al., 2011*). Also, *CD271*, *EYA2*, *EVC*, *MYF5*, *ZEB2*, and *TWIST1* in Cluster four were highly expressed in PAX7::EGFP+ cells. The expression of genes *MEF2C*, *MYH7*, *MYH3*, *MYOD1*, *TTN*, *DMD* (*Bentzinger et al., 2012*; *Mascarello et al., 2016*; *Stuart et al., 2016*; *Schiaffino et al., 2015*), and *RUNX1* in Cluster 10 gradually increased along muscle differentiation and reached peak levels at the late stage of myogenesis.

GO analysis of each cluster revealed significant enrichment of genes related to myogenesis, highlighting myogenic specification from hESCs into multinucleated myotube (*Figure 3C*). Analysis of Cluster 1 presented GO terms of Embryo Development (GO:0009790, p value, 8.37E-21), Segmentation (GO:0035282, 1.97E-09), Mesoderm Development (GO:0007498, 5.15E-07), and Somitogenesis (GO:0001756, 2.17E-06), which are related to somite generation. Myotube related genes indicated GO terms of Skeletal System Development (GO:0001501, 5.92E-07), Myofibril (GO:0030016, 7.69E-07), Muscle System Process (GO:0003012, 7.73E-07), and Muscle Contraction (GO:0006936, 1.64E-06) in Cluster 2. Cluster 3 and Cluster 4 mainly included GO terms involving gene expression and regulation of signaling; however, Skeletal Muscle Cell Differentiation (GO:0035914, 4.74E-05) and Muscle Organ Development (GO:0007517, 6.25E-05) were included only in Cluster 3. Cluster 10 involved Myofibril (GO:0030016, 1.67E-46), Sarcomere (GO:0030017, 2.13E-45), Muscle System Process (GO:0003012, 3.85E-35), and Muscle Contraction (GO:0006936, 4.96E-32), which indicated late myogenesis progress. This clustering analysis demonstrated that each cluster comprised unique gene expression profiles that may reveal new genetic regulators for in vitro human myogenesis.

## CD271 in Cluster 4 can mark PAX7::EGFP+ cells

One of the interesting transcripts in Cluster 4 was a cell surface marker, CD271 (*Figure 4A*). To test whether CD271 could be an alternative marker to isolate PAX7 expressing cells during in vitro myogenesis, we performed FACS analysis. Our data demonstrated that 92.55 ± 0.95% of PAX7::EGFP+ cells were positive for CD271 (mean ± SEM) (*Figure 4B* and *Figure 4—figure supplement 1A*). In post-sort analysis, 85.41 ± 3.24% of CD271$^{bright}$ cells expressed PAX7 based on antibody staining, whereas 3.41 ± 2.18% of CD271$^{low}$ cell population exhibited PAX7 immunoreactivity (*Figure 4C* and *Figure 4—figure supplement 1B*). To test the transcriptional enrichment of myogenic genes, we performed a post-sort qRT-PCR with CD271$^{bright}$ and CD271$^{low}$ populations and found that the levels of *PAX7* (fold changes, 27.81 ± 11.84), *MYF5* (43.85 ± 22.27), *MYOD1* (19.05 ± 10.68), and *MYOG* (16.33 ± 7.82) expression were significantly enriched in CD271$^{bright}$ cells compared to CD271$^{low}$ cells (*Figure 4D–E*). To determine the functionality of CD271$^{bright}$ cells, isolated CD271$^{bright}$ cells were cultured and passaged until passage three and differentiated to myotubes in each passage (*Figure 4—figure supplement 1C*). Quantification of the myotube marker staining (fusion index, please see Materials and methods) showed sustained fusion ability of the expanded CD271$^{bright}$ cells during in vitro expansion (for three weeks, while there are unfused cells), and the levels were comparable to those of the isolated PAX7::EGFP+ cells (*Figure 4F* and *Figure 4—figure supplement 1D–F*). While CD271 has been reported as one of the key factors during muscle development (*Reddypalli et al., 2005*), our data indicated that CD271 can mark PAX7 expressing putative skeletal muscle stem/progenitor population, and purified CD271$^{bright}$ cells can be expanded without losing robust myogenic capability.

## RUNX1 is expressed in subsets of myoblasts and multinucleated myotubes

The expression pattern of genes in Cluster 10 gradually increased during muscle differentiation, suggesting that the transcription factors in Cluster 10 can be new markers for the multinucleated myotube stage. For example, the *RUNX1* gene in Cluster 10 has been identified as a key transcriptional regulator for cell differentiation (*Zhu et al., 1994*) and demonstrated to be expressed in skeletal muscle (*Wang et al., 2005*). However, it is unknown in which cellular stage RUNX1 is expressed

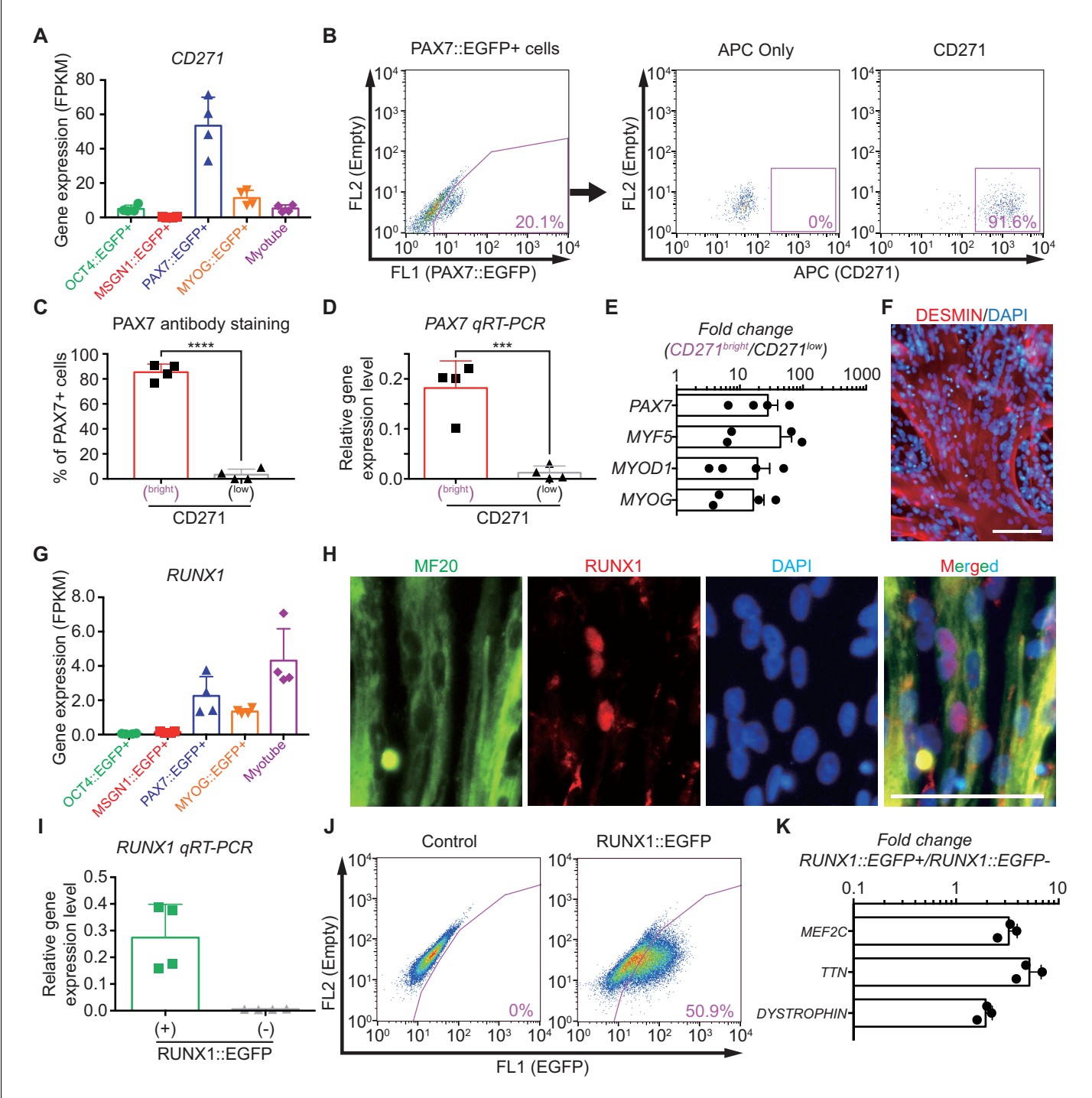

**Figure 4.** Characterization of CD271 and RUNX1 expression during in vitro myogenesis in hPSCs. (**A**) *CD271* expression profile with RNA-seq data in five different cell populations during muscle specification. (**B**) FACS plots of CD271 in PAX7::EGFP+ cell population at day 30 of skeletal muscle specification. (**C**) The percentages of PAX7+ cells in CD271^bright cells and CD271^low cells at 4 hr after cell sorting (****, *p* value < 0.0001; unpaired t-test). (**D**) *PAX7* gene expression levels of CD271^bright/CD271^low populations (***, p value < 0.001; unpaired t-test). (**E**) Fold changes of expression levels with myogenic-specific marker genes, *PAX7*, *MYF5*, *MYOD1* and *MYOG* in the CD271^bright/CD271^low populations. (**F**) Myotube formation ability of CD271^bright cells confirmed by DESMIN expression. (**G**) *RUNX1* expression profile with RNA-seq data in five different cell populations during muscle differentiation. (**H**) Co-localization of RUNX1 and MF20 in differentiating muscle cells at day 30. (bars, 100 μm) (**I**) The validation of reporting ability of the RUNX1::EGFP reporter line by measuring RUNX1 levels in both RUNX1::EGFP+ and RUNX1::EGFP- cell populations. (**J**) FACS plots for EGFP

*Figure 4 continued on next page*

*Figure 4 continued*

reporting RUNX1 expression in RUNX1::EGFP reporter line at day 35 of in vitro myogenesis. (**K**) Myotube related gene expression levels in both RUNX1::EGFP+ and RUNX1::EGFP- cell populations.

The online version of this article includes the following figure supplement(s) for figure 4:

**Figure supplement 1.** Stage-specific gene expression patterns in selected clusters.

during skeletal muscle final differentiation. RUNX1 is particularly interesting as it is clustered in Cluster 10, enriched in the Myotube group (*Figure 4G*). Antibody staining data demonstrated that immunoreactivity of RUNX1 is found in some multinucleated myotubes and myoblasts; however, we could not find any co-localization between PAX7 and RUNX1 antibody staining (*Figure 4H* and *Figure 4—figure supplement 1G–H*). Next, to determine the expression levels of RUNX1 in our skeletal muscle specification process (*Choi et al., 2016*), we generated a RUNX1::EGFP reporter hiPSC line that marks the expression of all known *RUNX1* isoforms (*Figure 4—figure supplement 1I–J*). In post-sort analysis, the RUNX1::EGFP+ population exhibited a significantly higher gene enrichment rate (p value, 0.005) of *RUNX1* than RUNX1::EGFP- cells (*Figure 4I*), confirming the reporting ability of our RUNX1::EGFP reporter line. At the late stage of skeletal muscle specification (days 30–35), RUNX1::EGFP expression levels were examined by FACS analysis and 44.90 ± 3.23% of total cells showed RUNX1::EGFP expression (mean ± SEM) (*Figure 4J*). We found significant transcriptional enrichment of myotube marker genes, *MEF2C* (fold changes, 3.26 ± 0.40), *TTN* (5.19 ± 0.90), and *DMD* (1.94 ± 0.18) (*Figure 4K*). As our differentiation protocol showed highly efficient generation of skeletal muscle cells with over 85% of MF20+ cells as shown in *Figure 1C*, myogenic marker genes were already enriched in the RUNX1::EGFP+ cells and RUNX1::EGFP- cells. Furthermore, additional qRT-PCR analysis with primer set of other lineage markers, including *GATA2* (endodermal), *FOXA2* (endodermal), *SOX17* (mesodermal), and *AGGRECAN* (mesenchymal) was confirmed that RUNX1::EGFP+ cells are not mixed with mesodermal, endodermal and mesenchymal cells (*Figure 4—figure supplement 1K*). These data demonstrate that RUNX1 is expressed during myogenic final differentiation and the RUNX1::EGFP+ cells are mostly skeletal muscle lineage cells.

## The role of TWIST1 in specification, maintenance and differentiation of human PAX7::EGFP+ cells derived from hPSCs

Genes in Cluster 4 show transcriptional activity only in PAX7::EGFP+ cells, suggesting that the transcription factors in Cluster 4 could have crucial roles in the biology of putative human skeletal muscle stem/progenitor cells derived from hPSCs. Among transcription factors in Cluster 4, TWIST1 is one of the interesting genes because TWIST2, an analog of TWIST, was reported as a key transcription factor in adult muscle progenitor cells in adult mouse (*Liu et al., 2017*). Considering the species differences between human and mouse and developmental stages between embryonic/fetal and postnatal stages, it will be interesting to investigate the roles of TWIST1 in in vitro human myogenesis, in particular PAX7 expressing cells. To investigate the function of the TWIST1 gene in Cluster 4 (*Figure 5A*), we generated three TWIST1 homozygote knock-out (KO) clones of the PAX7::EGFP reporter hESC line using the CRISPR/Cas9 System (*Figure 5B*). KO clone number one (#1) contained a deletion of 55 base pairs (bps) in one allele and 26 bps on the other allele including the start codon. KO Clone #3 had deletions of 9 bps on both alleles resulting in removal of the start codon of the TWIST1 gene. KO clone #7 had an insertion of one base pair (Adenosine, A) after the start codon and deletion of one base pair (A) after the start codon on the other allele, which induced frame-shifts on both alleles. Myogenic cells at day 30 derived from these KO clones did not express TWIST1 protein based on western blot experiment and antibody staining (*Figure 5—figure supplement 1A*) (TWIST1 was not detected in any of 1810 KO cells from three different clones), while TWIST1-positive cells were found in wild type (WT) (23.68 ± 2.25%, mean ± SEM).

To investigate the roles of TWIST1, we focused on three different aspects: specification of PAX7+ cells, maintenance of PAX7 expression in putative skeletal muscle stem/progenitor cells, and terminal differentiation for the generation of multinucleated myotubes. First, the TWIST1 KO clones in the PAX7::EGFP reporter line were subjected to our myogenic specification protocol (*Figure 5C*). The levels of PAX7 expression in TWIST1 KO cells were comparable to that of the WT clones between day 25 and day 30 (9.29 ± 0.68% vs. 9.40 ± 1.35, *Figure 5D* and *Figure 5—figure supplement 1B–*

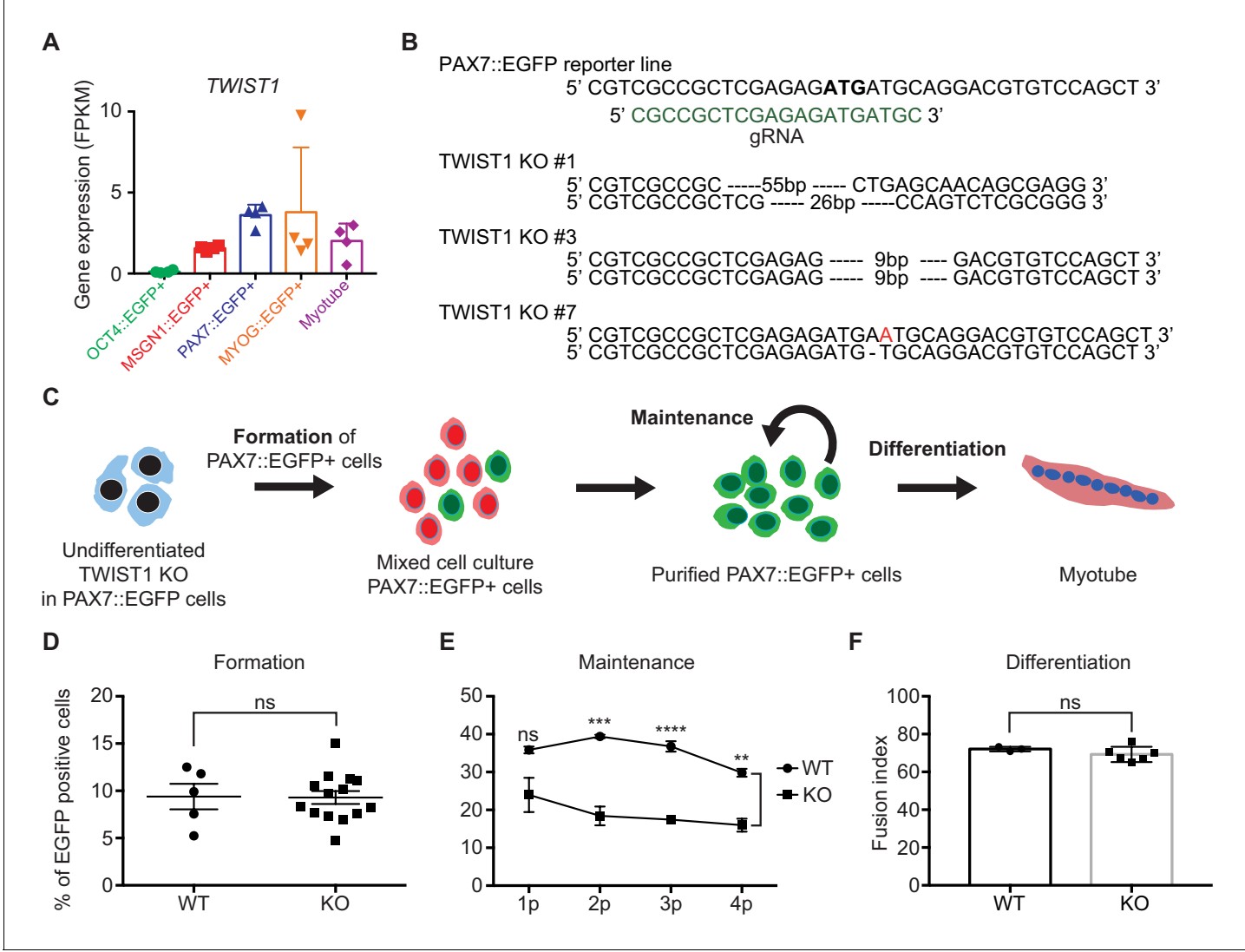

**Figure 5.** Establishment of TWIST1 knock-out (KO) lines in PAX7::EGFP reporter line and functional analysis of TWIST1 during human in vitro myogenesis. (**A**) *TWIST1* gene expression pattern of RNA-seq data in five different cell populations during muscle differentiation. (**B**) The position of genomic DNA and the sequence of gRNA for KO of the TWIST1 gene and the clone numbers with deleted sequences. (**C**) The experimental scheme of skeletal muscle specification, purification and differentiation. (**D**) The formation of PAX7::EGFP+ cells in WT and TWIST1 KO lines. (**E**) The maintenance of PAX7::EGFP+ cell population in WT and TWIST1 KO lines during PAX7::EGFP+ cell expansion. (**F**) The ability of multinucleated myotube formation in WT and TWIST1 KO lines.

The online version of this article includes the following figure supplement(s) for figure 5:

**Figure supplement 1.** The characterization of TWIST1 KO lines in PAX7::EGFP reporter line.

**Figure supplement 2.** Characterization of TWIST1 KO PAX7::EGFP+ cells.

*C*), demonstrating that TWIST1 KO does not affect the formation of PAX7+ skeletal muscle stem/ progenitor cells during in vitro human skeletal muscle generation. Next, to see if TWIST1 has a role in cell proliferation of PAX7::EGFP+ cells, we tested their proliferation abilities during in vitro expansion. We found 69.88 ± 3.49% of Ki67+ cells in TWIST1 KO PAX7::EGFP+ cells, which was comparable to that of WT clones, indicating that TWIST1 did not affect cell proliferation capacity (*Figure 5— figure supplement 1D*). The level of apoptosis (examined using Annexin V analysis) in TWIST1 KO PAX7::EGFP+ cells was similar to that of WT clones (*Figure 5—figure supplement 1E*). To determine the functionality of TWIST1 KO PAX7::EGFP+ cells, isolated PAX7::EGFP+ cells were cultured until passage four and differentiated to myotubes in each passage (*Figure 5F* and *Figure 5—figure supplement 1F*). We found that there was no difference in the fusion index (MF20 antibody staining)

between TWIST1 KO and WT PAX7::EGFP+ cells, indicating that *TWIST1* deletion does not cause any detectable defects in multinucleated myotube formation. Interestingly, there are aberrant transcriptional patterns of *PAX3*, *PAX7*, *MYF5*, and *MYOD1* during the myogenic specification (*Figure 5—figure supplement 2A*), although the levels of PAX7::EGFP+ cells in WT and TWIST1 KO clones (at Day 25 to Day 30 of myogenic specification) are comparable (*Figure 5D* and *Figure 5—figure supplement 2B–C*). To test the role of TWIST1 in maintenance of PAX7 expression in the isolated PAX7::EGFP+ cells, we performed weekly FACS analysis during cell expansion for a month. While both WT and KO PAX7::EGFP+ cells showed comparable levels of PAX7 expression at the early passages, the percentages of PAX7::EGFP+ cells in late passages (at passages 2 to 4) were significantly lower in TWIST1 KO PAX7::EGFP+ cells (15.98 ± 1.70% at passage 4) compared to the percentage of PAX7::EGFP+ cells in WT (29.80 ± 1.04% at passage 4) (p values, 0.0983, 0.0008, <0.0001, and 0.0012 at passages 1, 2, 3, and 4, respectively) (*Figure 5E* and *Figure 5—figure supplement 2B*). Furthermore, we could not find any significantly difference between the WT and the KO clones in terms of cell morphology, including cellular size and length, proliferation rates, and the proportions of PAX7+/MYOG+ cells (*Figure 5—figure supplement 2C–G*). These results revealed that TWIST1 deletion has a significant effect on maintaining PAX7::EGFP expression in putative human skeletal muscle stem/progenitor cells, but loss of TWIST1 has no effect on proliferation or apoptosis during in vitro expansion. Taken together, our results demonstrated that TWIST1 might have an important role in the maintenance of PAX7 expression in putative human skeletal muscle stem/progenitor cells.

In order to understand the underlying mechanism of TWIST1's role in in vitro human myogenesis at the genome-wide level, we carried out RNA-seq analysis for the purified cells from TWIST1 Knock-Out (KO) PAX7::EGFP+ cells. PCA showed that TWIST1 KO PAX7::EGFP+ cells were distinct from the other five cell populations in terms of genetic distance and relatedness (*Figure 6—figure supplement 1A* and *Supplementary file 4*, related to *Figure 2C*). To identify differential gene expression between TWIST1 KO PAX7::EGFP+ cells and WT PAX7::EGFP+ cells, we performed unsupervised hierarchical clustering analysis (*Figure 6A* and *Supplementary file 4*). The heatmap showed 643 differentially expressed genes between WT PAX7::EGFP+ cells and TWIST1 KO PAX7::EGFP+ cells. To investigate the similarities and dissimilarities in the global gene expression between WT PAX7::EGFP+ cells and TWIST1 KO PAX7::EGFP+ cells, a Venn diagram was plotted (*Figure 6B* and *Supplementary file 4*). Out of 4317 genes highly upregulated over OCT4::EGFP+ cell populations, WT PAX7::EGFP+ cells and TWIST1 KO PAX7::EGFP+ cells had 1996 shared genes with GO terms of Extracellular Matrix Organization (GO:0030198), Muscle Filament Sliding (GO:0030049), and Collagen Catabolic Process (GO:0030574). The 1736 genes upregulated in TWIST1 KO PAX7::EGFP+ cells were classified to GO terms such as Translational Initiation (GO:0006413), SRP-dependent cotranslational protein targeting to membrane (GO:0006614), and Viral Transcription (GO:0019083).

In order to better understand the important biological features between these two populations, we selected upregulated transcription factors in PAX7::EGFP+ cells and TWIST1 KO PAX7::EGFP+ cells (*Figure 6C* and *Supplementary file 4*). The upregulated TFs in WT PAX7::EGFP+ cells included *HES6*, *EGR2*, *YBX3*, and *TWIST1*. On the other hand, *YBX1*, *ZEB1*, *TCF12*, *SSRP1*, *NFE2L1*, *EBF3*, *BARX2*, *CTCF*, *FOXM1*, and *SMARCC2* were upregulated TFs in TWIST1 KO PAX7::EGFP+ cells. Next, we performed statistical Gene Ontology (GO) analysis between PAX7::EGFP+ cells and TWIST1 KO PAX7::EGFP+ cells (*Figure 6D* and *Supplementary file 4*). The GO terms indicated Response to Endoplasmic Reticulum Stress (GO:0034976, p value, 5.95E-04), Skeletal Muscle Tissue Development (GO:0007519, 9.35E-03), Skeletal Muscle Organ Development (GO:0060538, 1.06E-02), Muscle Organ Development (GO:0007517, 1.40E-02), and Transmembrane Transport (GO:0055085, 1.76E-02) in WT PAX7::EGFP+ cells. GO terms of Regulation of mRNA Processing (GO:0050684, 7.41E-05), Cell Division (GO:0051301, 1.96E-04), RNA Processing (GO:0006396, 1.83E-03), ER to Golgi Vesicle-mediated Transport (GO:0006888, 6.34E-03), and Wnt Signaling Pathway (GO:0016055, 7.63E-03) were enriched in TWIST1 KO PAX7::EGFP+ cells, suggesting that TWIST1 KO can affect a wide range of biological events.

It has been demonstrated that Twist1 is directly bound to *Pax7* locus in non-skeletal muscle tissues based on Chip-seq analysis by other groups (*Chang et al., 2015*; *Lee et al., 2014*), which might explain the loss of PAX7::EGFP expression in TWIST KO PAX7::EGFP cells. We focused on the WNT signal pathway among the prominent GO terms, because the Wnt signaling pathway plays an

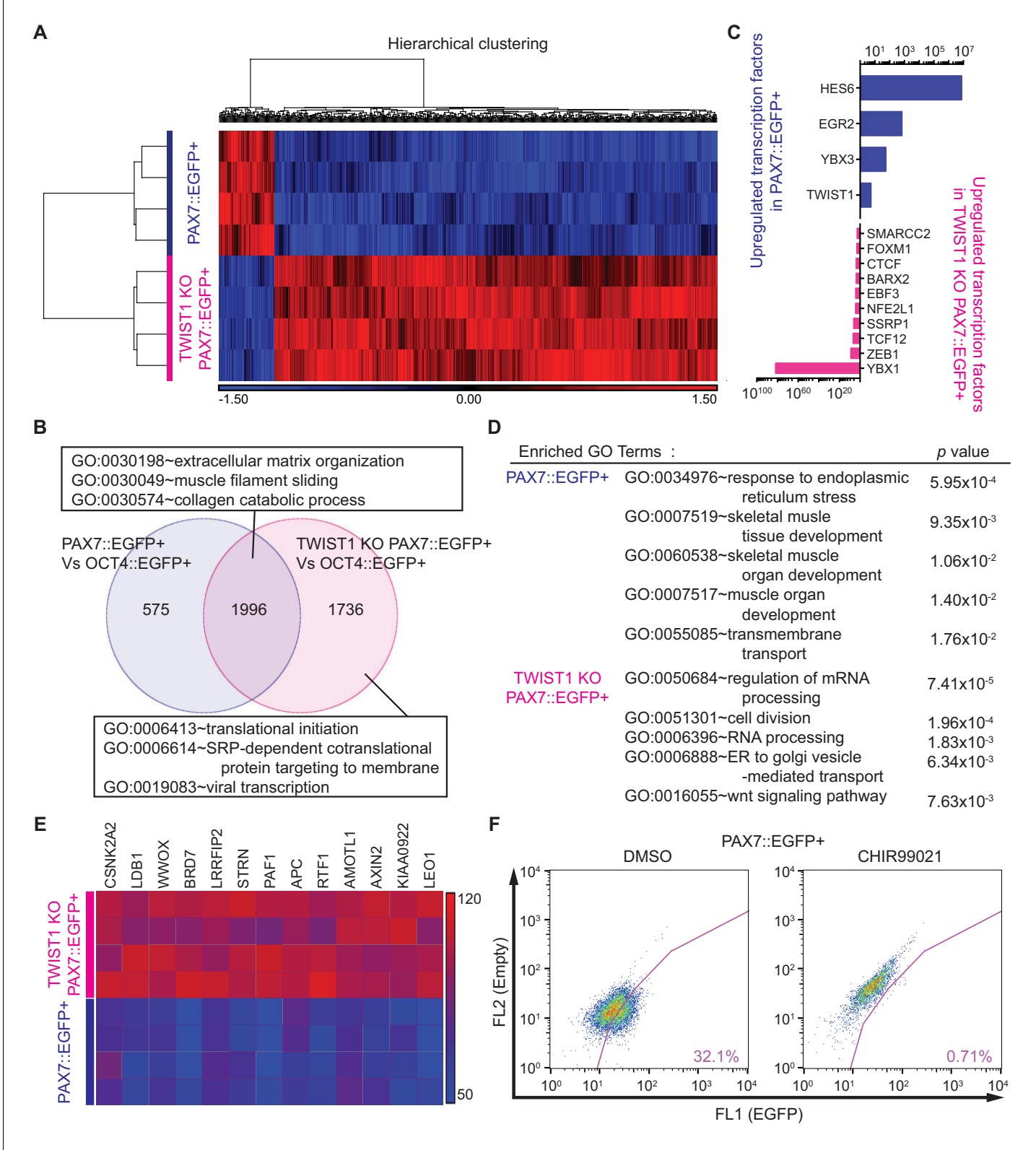

**Figure 6.** Whole transcriptome analysis between PAX7::EGFP+ cells and TWIST1 KO PAX7::EGFP+ cells. (**A**) Heatmap and hierarchical clustering of RNA-seq data for PAX7::EGFP+ cells and TWIST1 KO PAX7::EGFP+ cells. (**B**) Venn diagram of the upregulated genes and their GO terms in PAX7:: EGFP+ cells and TWIST1 KO PAX7::EGFP+ cells compared to OCT4::EGFP+ cells. (**C**) List of upregulated TFs in PAX7::EGFP+ cells and TWIST1 KO PAX7::EGFP+ cells compared to each other. (**D**) Enriched Gene ontology (GO) terms with whole transcriptome that had high p values between PAX7::

*Figure 6 continued on next page*

*Figure 6 continued*

EGFP+ cells and TWIST1 KO PAX7::EGFP+ cells from RNA-seq data. (E) Heatmap of Wnt signaling downstream target gene expression patterns in PAX7::EGFP+ cells and TWIST1 KO PAX7::EGFP+ cells. (F) FACS plots of EGFP percentage with Wnt activator (CHIR99021) in PAX7::EGFP+ cells during in vitro cell proliferation.

The online version of this article includes the following figure supplement(s) for figure 6:

**Figure supplement 1.** Transcriptome analysis between PAX7::EGFP+ cells and TWIST1 KO PAX7::EGFP+ cells.
**Figure supplement 2.** Stage-specific modules of RNA-seq data from five different cell types with weighted gene co-expression network analysis.

important role in maintaining stemness of adult satellite cells (*Girardi and Le Grand, 2018*; *Jones et al., 2015*; *Biressi et al., 2014*; *Bentzinger et al., 2014*; *Bentzinger et al., 2013*). Also, RNA-seq data indicated that direct downstream target genes of WNT signaling, such as *CSNK2A2*, *LDB1*, *WWOX*, *BRD7*, *LRRFIP2*, *STRN*, *PAF1*, *APC*, *RTF1*, *AMOTL1*, *AXIN2*, *KIAA0922*, and *LEO1* were highly upregulated in TWIST1 KO PAX7::EGFP+ cells (*Figure 6E*). We treated WT PAX7::EGFP + cells with the WNT activator (CHIR99021), which decreased the number of PAX7::EGFP+ cells during in vitro expansion in a dose-dependent manner (*Figure 6F* and *Figure 6—figure supplement 1B*) and mimicked the phenotypes of TWIST1 KO PAX7::EGFP cells. On the other hand, treatment of TWIST1 KO PAX7::EGFP cells with the WNT inhibitor (XAV939) chemical compound reversed the TWIST1 KO phenotype, but we could not find any statistically significant effects on maintenance of PAX7::EGFP expression (*Figure 6—figure supplement 1C–D*). These data demonstrated that TWIST1 is crucial for in vitro expansion of PAX7::EGFP+ cells, and it is associated with the WNT signaling pathway.

## Discussion

Here, we present a comprehensive analysis for the transcriptome of hPSC-derived myogenic specification. By utilizing multiple reporting hPSC lines, we reconstructed transcriptionally defined stages of human in vitro myogenesis and uncovered sets of novel genes including transcription factors, followed by validation of a new gene, *TWIST1*. Our data provide insights into the dynamics of the transcriptome during in vitro myogenesis and demonstrate the ability to study significant mechanisms in a stage specific manner during embryonic myogenesis.

As shown in stem cell research to isolate specific sub-populations during differentiation processes (*Ganat et al., 2012*; *Tchieu et al., 2017*), genetic reporter stem cell lines are important tools to study transcriptional regulation and functions in various cell fate determination processes, and they are useful to understand the dynamic fluctuations of expression of key transcription factors. In this study, we successfully utilized six individual 'knock-in' genetic reporter hPSC lines targeting key myogenic specification genes. These new reporter cell lines allowed us to prospectively identify and purify specific cellular populations with high purity during in vitro human myogenesis, leading us to delineate a comprehensive transcriptional landscape of human skeletal muscle development.

Early embryonic myogenesis has been investigated in animal models, and key transcription factors, such as *PAX7* and *MYOG*, have been discovered (*Bentzinger et al., 2012*; *Kassar-Duchossoy et al., 2005*; *Parker et al., 2003*); however, the transcriptional similarity and disparity between PAX7 expressing cells (muscle stem/progenitor cells) and MYOG expressing cells (myoblast cells) has not been well characterized. By using PAX7::EGFP and MYOG::EGFP genetic reporter hPSC lines, we were able to capture two distinct but converged cellular populations during human in vitro myogenesis. We believe that the myogenic specification from human pluripotent stem cells are not synchronized events, as we see that gradual increase of PAX7::EGFP expression during the time course. Also the in vitro condition may not be favorable to the maintenance of PAX7 expression, which forces the PAX7::EGFP+ cells to be differentiated into MYOG::EGFP+ cells. Therefore, we believe that in vitro culture condition (favorable for differentiation) and the transcriptional heterogeneity of PAX7::EGFP+ cells are responsible for the slightly increased levels of MYOG::EGFP+ cells in early myogenic specification stage (Day 20–22) in the *Figure 1D*. However, our data do not necessarily conclude that MYOG+ cells are not derived from PAX7::EGFP+ cells. To understand the transcriptional association and distinctive characteristics between PAX7::EGFP+ cells and MYOG::EGFP + cells, we compared these two populations to OCT4::EGFP+ cells to tease out detailed transcriptional similarities and examine differential gene expression profiles (*Figure 2*). Our data indicate that

over-represented GO terms of the enriched gene set in PAX7::EGFP+ cells involve DNA binding and/or signaling regulation. For instance, the list of enriched transcription factors in PAX7::EGFP+ cells, but not in MYOG::EGFP+ cells, include genes of the JUN signaling pathway family, such as *JUN*, *JUNB* and *FOS* genes, which are known to prevent the activation of MYOG (*Li et al., 1992*). ID3, a well-known downstream target gene of PAX7 as reported in mouse skeletal muscle stem cells (*Kumar et al., 2009*; *van Velthoven et al., 2017*), is also expressed in the PAX7::EGFP+ cells. The HMG family of DNA binding coding genes, including HMGB1, HMGB2, HMGA1, and HMGB3, are also enriched in the PAX7::EGFP+ population, suggesting their potential roles in skeletal muscle stem/progenitor properties. The HMGB1 gene is heavily involved in regulation of proliferation and differentiation in other stem cell populations (*Meng et al., 2018*; *Wang et al., 2014*).

On the other hand, the list of upregulated transcription factors in MYOG::EGFP+ cells compared to PAX7::EGFP+ cells contains previously known myoblast marker genes, such as MYOG, MEF2C, and MYOD1 (*Arnold and Braun, 1996*). Also, increased FOXO1 and FOXO4 transcription levels in MYOG::EGFP+ cells might be related to the proliferation ability of MYOG::EGFP+ cells, since the FOXO family contributes to proliferation of myoblasts through the AKT signaling pathway (*Gross et al., 2008*; *Xu et al., 2017*). Similarly, a list of genes coding for the ERK signaling pathway proteins, such as ATF4, TEAD4, and SNAI1, are enriched in the MYOG::EGFP+ population, suggesting that properties of myoblasts are related to the ERK signaling pathway as previously shown in animal studies (*Ayuso et al., 2016*; *Benhaddou et al., 2012*; *Horak et al., 2016*).

The clustering analysis of gene expression data is beneficial for identification of novel genes, finding specialized functions associated with continuous biological events, and discovering transcriptionally distinct properties of defined stage-specific cell populations. K-mean clustering was applied to group highly correlated gene expression patterns during hPSC-based in vitro myogenesis and classified the genes into 10 clusters. This approach gave us new insights that divergent gene expression patterns contribute to different stages during in vitro myogenesis. Cluster 10 involves a list of genes that have gradually increased expression patterns during myogenesis. Although most of the genes in Cluster 10 have roles in myotube formation during skeletal muscle generation, some of them have functions in non-skeletal muscle lineages. For example, Cluster 10 includes RUNX1, which has critical roles in hematopoiesis and leukemogenesis (*Ichikawa et al., 2013*). Our data indicate that the *RUNX1* gene is transcriptionally and translationally active in our human myotube samples, but not in myoblasts based on our data with the RUNX1::EGFP reporter cell line. While the TMEM8C (MYO-MAKER) gene involved in Cluster 10 was studied as a membrane activator of myoblast fusion (*Millay et al., 2013*), Cluster 3 includes the MYMX (MYOMIXER) gene, which was reported as a key factor to induce myoblast fusion (*Bi et al., 2017*). MYMX and TMEM8C are known to be expressed in myoblasts and induce myotube formation, and a recent study reported that these genes contribute to distinct steps in myotube formation processes (*Leikina et al., 2018*). Another interesting cluster is Cluster 4, which includes a gene list related to muscle stem/progenitor markers, such as mouse muscle stem cell markers including PAX7, PAX3, SYND3, and BARX2 (*Bentzinger et al., 2012*; *Yin et al., 2013*; *Magli and Perlingeiro, 2017*). The GO terms represented in Cluster 4 suggest that genes belonging to muscle development categories and signaling pathways are involved in biological functions that are related to the properties of skeletal muscle stem/precursor cells. Interestingly, the CNBP gene, a culprit gene that causes myotonic dystrophy 2 (MD2) (*Meola and Cardani, 2015*), belongs to Cluster 4, enriched in the PAX7::EGFP+ cell population. In an attempt to further analyze the stage-specific transcriptome regulation during human myogenesis, we applied weighted gene co-expression network analysis (WGCNA) in five different cell types (*Figure 6—figure supplement 2*). WGCNA with stages-specific transcriptomes allowed us to define modules of genes that are continuously coregulated and to study their stage-specific variation during skeletal muscle differentiation. By including all expressed genes with expression variation, we identified a total of 92 modules defined as groups of genes coordinately expressed across 20 samples. These data imply that cellular dysfunction might be initiated even in the muscle stem/precursor cell stage, which could provide new insights to study MD2 pathogenesis.

Similar patterns can be found in DYSTROPHIN, the culprit gene for Duchenne muscular dystrophy (DMD). Recently, the expression of DYSTROPHIN has been studied in mouse satellite cells, demonstrating that DYSTROPHIN is responsible for cell polarity and asymmetric division of satellite cells (*Dumont et al., 2015*; *Chang et al., 2018*). Previously, DYSTROPHIN had been known as an important 'linkage' protein connecting skeletal muscle fibers of cytoskeleton to the extracellular matrix,

and DYSTROPHIN's functional failure results in severe skeletal muscle degeneration, which is the etiology of DMD (*Hoffman et al., 1987*). Although DYSTROPHIN is clustered to Cluster 10, we also found minute but detectable levels of DYSTROPHIN expression in PAX7::EGFP+ cells (*Figure 3—figure supplement 1B*). To confirm our transcriptional analysis data, we performed western blot with DYSTROPHIN antibody (MANDYS1 clone) in isolated PAX7::EGFP+ cells and demonstrated that DYSTROPHIN is also transcriptionally and translationally expressed in hESC-derived PAX7::EGFP+ cells (*Figure 3—figure supplement 1C*).

CD271 is important for normal development of muscle from mutant mice (*Reddypalli et al., 2005*). While prior studies were performed using surface markers for the isolation of myoblast (*Hicks et al., 2018*; *Sakai-Takemura et al., 2018*), our studies were focused more PAX7+ skeletal muscle stem/progenitor cells during in vitro human skeletal muscle generation. Our data indicate that CD271$^{bright}$ population overlaps with PAX7::EGFP+ cells in FACS analysis (*Figure 4*) and shows efficient myotube formation ability according to fusion index during in vitro culture. With the advantages of using a surface marker, CD271 can be a substitute for the PAX7::EGFP reporting system, and it could be applied to other cell types for skeletal muscle disease modeling without tedious and time-consuming steps for generating reporter lines. In addition, other neurotrophin receptors, TRKA and TRKB, were clustered to Cluster 4, which suggests that neurotrophins and their signaling pathways could be one of the key regulatory mechanisms in PAX7 expressing skeletal muscle stem/progenitor cells as shown in previous rodent studies (*Griesbeck et al., 1995*; *Clow and Jasmin, 2010*). Collectively, our data suggest that neurotrophin receptor genes, such as CD271, TRKA and TRKB, might have direct/indirect contributions in the process of skeletal muscle development.

TWIST1 has similarity to TWIST2, and TWIST2 was reported as a key transcription factor in adult muscle progenitor cells in mouse (*Liu et al., 2017*). However, RNA expression of TWIST2 was barely detected in our RNA-seq results, and TWIST1 is contained in Cluster four along with the PAX7 gene expression pattern. It has been demonstrated that *Twist1* is actively expressed in mouse adult satellite cells, based on Chip-seq study, marked by H3K4me3 (*Woodhouse et al., 2013*), but its function has not been clarified. Although TWIST1 has been known for its role cancer progression (*Parajuli et al., 2018*), two other studies with fly and axolotl system demonstrate that TWIST1 is implicated with skeletal muscle regeneration (*Kragl et al., 2013*; *Baylies and Bate, 1996*). In axolotl, as the limb bud elongates, Twist1 expression was found sub-epidermally toward the distal portion of the limb bud and co-localized with some, but not all, of MYF5 and PAX7 expressing muscle cells (*Kragl et al., 2013*). In*Drosophila* Twist (Twi) regulates mesodermal differentiation and propels a specific subset of mesodermal cells into somatic myogenesis (*Baylies and Bate, 1996*). However, TWIST1 expression in human myogenic development and its relevance to genetic diseases has not been studied until now. Saethre-Chotzen syndrome (OMIM # 101400) is caused by *TWIST1* mutations, and it is an autosomal dominant craniosynostosis syndrome with uni- or bi-lateral coronal synostosis and mild limb deformities (Musculoskeletal Diseases) (*Sharma et al., 2013*; *Murphy et al., 2018*). While the neural crest-related defects in Saethre-Chotzen syndrome patients have been studied, the underlying mechanism of musculoskeletal symptoms is unknown. Importantly, our results show that TWIST1 deletion (three different mutations in each clone) affects the maintenance of PAX7 expression levels in human PAX7::EGFP cells during in vitro expansion. These data suggest that TWIST1 plays a role in putative skeletal muscle stem/progenitor cells and provide experimental evidence to begin to understand pathogenic events responsible for the musculoskeletal symptoms in Saethre-Chotzen syndrome patients. In future studies, it will be important to explore the stage-specific role of TWIST1 and characterize more detailed mechanisms of musculoskeletal symptoms in Saethre-Chotzen syndrome.

The WNT signaling pathway has been reported to play various roles in skeletal muscle development and skeletal muscle stem cell biology (*Girardi and Le Grand, 2018*; *Jones et al., 2015*). In addition, it has been shown that Twist1 expression is induced by canonical Wnt signaling and has an important role in inhibiting myotube formation by sequestering DNA binding of MYOD1 (*Miraoui and Marie, 2010*). Our RNA-seq data show that the lack of TWIST1 protein in human PAX7::EGFP+ cells significantly upregulated a set of genes related to WNT signaling pathways, and treatment with glycogen synthetase kinase 3 (GSK3) inhibitor, a WNT activator, decreased PAX7 gene expression (*Figure 6—figure supplement 1B*). Moreover, modulating the WNT signaling pathway did not rescue the TWIST1 KO phenotype (*Figure 6—figure supplement 1D*). These data

suggest that TWIST1 could be a mediator to link the canonical WNT signaling pathway to PAX7 gene expression and likely maintenance of putative skeletal muscle stem/progenitor cell identity.

Overall, our study depicts a transcriptional landscape of in vitro myogenesis from hPSCs, which is a valuable platform to deconstruct/reconstruct the dynamic transcriptional activities of multiple stages during human skeletal muscle specification. Our clustering analysis of RNA-seq data will be useful for understanding global gene expression during human in vitro myogenesis and an ideal reference for mining new myogenic regulator genes. In depth understanding of transcriptional dynamics of human myogenesis and identification of a cell intrinsic requirement for maintenance of PAX7 expression should facilitate development of novel strategies enhancing functional repair of many degenerative muscle conditions. We believe that these data will be beneficial for future applications, such as skeletal muscle related disease modeling and genetic engineering for cell replacement therapy.

## Materials and methods

### Key resources table

| Reagent type (species) or resource | Degignation | Source or reference | Identifiers | Additional information |
|---|---|---|---|---|
| Cell line (*Homo-sapiens*) | H9 Human ES | WiCell | WAe009-A | |
| Antibody | α-ACTININ (Sarcomeric) (mouse monoclonal) | Sigma-Aldrich | Cat#: A7811 RRID: AB_476766 | (1:1000) |
| Antibody | AP2 (rabbit polyclonal) | DSHB | Cat#: 3B5 RRID: AB_528084 | (1:500) |
| Antibody | CD271 (mouse monoclonal) | Advanced Targeting Systems | Cat#: AB-N07 RRID: AB_171797 | (1:1000) |
| Antibody | DESMIN (mouse monoclonal) | DAKO | Cat#: M0760 RRID: AB_2335684 | (1:1000) |
| Antibody | DYSTROPHIN (mouse monoclonal) | DSHB | Cat#: MANDYS1(3B7) RRID:AB_528206 | (1:1000) |
| Antibody | MF20 (mouse monoclonal) | DSHB | Cat#: MF20 RRID: AB_2147781 | (1:1000) |
| Antibody | MYOG (mouse monoclonal) | DSHB | Cat#: F5D RRID: AB_2146602 | (1:1000) |
| Antibody | NANOG (rabbit polyclonal) | Cell Signaling | Cat#: 3580 RRID:AB_2150399 | (1:1000) |
| Antibody | OCT4 (mouse monoclonal) | SANTA CRUZ | Cat#: sc-5279 RRID:AB_628051 | (1:1000) |
| Antibody | PAX7 (mouse monoclonal) | DSHB | Cat#: pax7 RRID:AB_528428 | (1:1000) |
| Antibody | RUNX1 (rabbit polyclonal) | abcam | Cat#: ab23980 RRID:AB_2200834 | (1:100) |
| Antibody | SOX10 (mouse monoclonal) | SANTA CRUZ | Cat#: sc-17343 RRID:AB_2255319 | (1:1000) |
| Antibody | TRA1-81 (mouse monoclonal) | Cell Signaling | Cat#: 4745 RRID:AB_2119060 | (1:1000) |
| Antibody | TWIST1 (mouse monoclonal) | abcam | Cat#: ab175430 | (1:100) |
| Commercial assay or kit | Proteome Profiler Human Phospho-Kinase Array kit | R and D Systems | Cat#: ARY003B | |
| Other | DAPI | Invitrogen | D1306 | (1 ug/ml) |

### Cell culture and muscle specification

WA09 (H9) human embryonic stem cell line was used (purchased from WiCell, confirmed by STR by WiCell, mycoplasma free by Lee lab). Culturing H9 hESCs (WiCell) and muscle specification were

performed as previously described (*Choi et al., 2016*). Briefly, hESCs were maintained on mouse embryonic fibroblasts (MEFs, Applied StemCell) at 12,000 to 15,000 cells/cm$^2$, and the cells were fed daily and passaged weekly using either 6 U/mL Dispase or mechanical means. For muscle specification, hESCs were dissociated to single cells using Accutase for 20 min and then plated on a 1% Geltrex-coated dish at the proper density in the presence of conditioned N2 media containing 10 ng/ml of FGF-2 and 10 µM of Y-27632 (Cayman Chemical Co.). At ~70% confluence, N2 media with CHIR99021 (3 µM, Cayman Company) was added, and media was changed every other day. At day 4, N2 media with DAPT (10 µM, Cayman Company) was added until day 12. Cells were harvested at different time points as described. For the maintaining EGFP+/-cells after cell sorting, EGFP+/-cells were plated in 1% Geltrex-coated dish with N2 media containing 5% FBS, 10 ng/ml FGF-2 and 100 ng/ml FGF-8. Media were changed at every other day. For passaging, cells were dissociated with Accutase for 20 min, washed once with N2 media.

## Generation of genetic reporter hPSC lines

Genetically manipulated hPSC lines were generated using the CRISPR/Cas9 system as previously described (*Choi et al., 2016*). In brief, approximately 1.5 kbps of the homologous arm next stop codon of the targeted gene was amplified by genomic DNA PCR and cloned into a proper plasmid. The guide-RNAs were constructed as described by the manufacturer's manual (Addgene). Nucleofection for gene targeting was performed according to manufacturer's instruction (Lonza). Briefly, 1 µg donor vector, 1 µg guide RNA, and 1 µg Cas9 vector (Addgene plasmid #41815) were nucleofected into $2 \times 10^6$ dissociated hPSCs using the AMAXA nucleofector II. Nucleofected hPSCs were plated onto puromycin resistant MEFs (Applied StemCell), and cells were treated with puromycin for colony selection. Puromycin-resistant colonies were manually picked and expanded. EGFP expression was confirmed by FACS analysis after myogenic differentiation. For the RUNX1::EGFP reporter line, 5 µg donor vector, 2.5 µg ZFN1, and 2.5 µg ZFN2 from Sangomo were nucleofected into dissociated BC1 hiPSCs, which were generated from human adult bone marrow CD34+ cells (*Chou et al., 2011*; *Connelly et al., 2014*). Neomycin-resistant colonies were picked and confirmed by PCR assay.

## Transcription analysis and immunofluorescence

Total RNA was extracted from differentiating hPSC lines using TRIzol Reagent (Life Technologies), and 1 µg of RNA was reverse transcribed using the High-Capacity cDNA Reverse Transcription Kit (Applied Biosystems). The qRT-PCR mixtures were prepared with SYBR Green PCR Master Mix Universal (Kapa Biosystem), and reactions were performed using Eppendorf Realplex$^2$. The transcription levels were assessed by normalizing to GAPDH expression. For immunofluorescence, cells were fixed with 4% paraformaldehyde and permeabilized with 0.5% Triton X-100 in PBS solution. Primary antibodies were applied after blocking with 0.5% BSA solution. For imaging, the stained cells were visualized using Nikon Eclipse TE2000-E fluorescence microscopy. Fusion index was measured as previously described (*Choi et al., 2016*). Briefly, to determine myoblast fusion rates, cells were differentiated for 10 days and stained with MF20 antibody. Fusion index was calculated as the ratio of the number of nuclei inside MF20+ myotubes over the number of total nuclei in the image.

## FACS analysis and sorting

For flow cytometry, cells were dissociated with Accutase and treated with DNase for 20 min at 37℃. The cells were analyzed by BD FACS Calibur (Becton Dickinson), and data were visualized using FlowJo software (Tree Star Inc). For the MF20 FACS analysis, cells were dissociated with Accutase and MF20 was applied and incubated 20 min at 25℃. Secondary antibody was used for the detecting by BD FACS Calibur (Becton Dickinson). Cell isolation experiments were performed by BD ARIA II at the Johns Hopkins Flow Cytometry Core Facility.

## Global gene expression profile

RNA-seq libraries were constructed using Illumina TruSeq Stranded Total RNA RiboZero Gold sample Prep Kit (RS-122–2303) according to the manufacturer's protocol (Illumina). cDNA libraries using unique barcoded adapters from 20 samples were analyzed with next-generation multiplexed sequencing Illumina Hiseq 3000, which resulted in high-quality output, with a mean quality score (Q score)>30% and 90% of perfect index reads for all samples. After the sequencing run, the Illumina

Real Time Analysis (RTA) module was used to perform image analysis, followed by base calling and the BCL Converter (bcl2fastq v2.17.1.14) to generate FASTQ files that contain the sequence reads. Pair-end reads of cDNA sequences were aligned back to the human genome (UCSC hg19 from Illumina iGenome) by the spliced read mapper TopHat (v2.0.9). The sequencing depth was over 90 million (45 million paired-end) mappable sequencing reads. The alignment statistics and Q/C were achieved by SAMtools (v0.1.18) and RSeQC (v2.3.5) to calculate quality control metrics on the resulting aligned reads, which provided useful information on mappability, uniformity of gene body coverage, insert length distributions and junction annotation. The PCA plot and heatmap from the RNA-seq data were generated according to the manufacturer's instructions (Partek Genomics Suite, version 6.6).

### K-mean clustering
Normalized gene expression RNA-seq data sets of step wise specification from hPSCs to skeletal muscle cells were further grouped via a K-mean clustering algorithm. The algorithm split the data into 10 clusters of genes with similar expression patterns over all five different specific stages. The number of correct clusters was determined by measuring the average of intracluster and intercluster distances based on the similarity of genes to genes in its own cluster as compared to genes in other clusters.

### Statistical analysis
All statistical analyses were performed using the Graph Pad Prism software (version 6.0). The values were from at least three independent experiments with multiple replicates per each experiment, and they were reported as mean ± SEM. Comparisons among the groups were performed by either one-way ANOVA followed by Newman-Keuls test or an unpaired t-test. Statistical significance was assigned at $p < 0.05$.

### Data deposition
The RNA-seq data were deposited to NCBI (GSE129505).

## Acknowledgements
This work was supported by grants from the National Institutes of Health through R01NS093213, R01AR070751 (GL), the Maryland Stem Cell Research Funding (MSCRF; GL), the Muscular Dystrophy Association (MDA: GL), the FSH Society (GL), and the Global Research Development Center program from the Korea National Research Foundation (GL, S-H H). The authors acknowledge the Cytogenetic Core Facility (supported by the Eunice Kennedy Shriver National Institute of Child Health and Human Development of the National Institutes of Health under Award Number U54HD079123). We thank the Developmental Studies Hybridoma Bank for antibodies.

## Additional information

### Funding

| Funder | Grant reference number | Author |
|---|---|---|
| National Institute of Arthritis and Musculoskeletal and Skin Diseases | R01AR070751 | Gabsang Lee |
| Maryland Stem Cell Research Fund | 2017-MSCRFD-3941 | Gabsang Lee |
| National Institutes of Health | R01NS093213 | Gabsang Lee |
| Maryland Stem Cell Research Fund | 2017-MSCRFD-3941 | Gabsang Lee |
| Muscular Dystrophy Association | 381465 | Gabsang Lee |
| FSH Society | FSHS-82014-03 | Gabsang Lee |

| National Research Foundation of Korea | 2017K1A4A3014959 | SangHwan Hyun Gabsang Lee |

The funders (NIAMS and MSCRF) had no role in study design, data collection and interpretation, or the decision to submit the work for publication.

## Author contributions

In Young Choi, Data curation, Formal analysis, Validation, Investigation, Visualization, Methodology, Writing - original draft, Writing - review and editing; Hotae Lim, Conceptualization, Data curation, Formal analysis, Validation, Investigation, Visualization, Methodology, Writing - original draft, Writing - review and editing; Hyeon Jin Cho, Software, Formal analysis, Investigation, Visualization; Yohan Oh, Hao Bai, Resources, Methodology; Bin-Kuan Chou, Resources; Linzhao Cheng, Resources, Supervision; Yong Jun Kim, SangHwan Hyun, Supervision, Investigation; Hyesoo Kim, Data curation, Supervision, Investigation, Visualization, Writing - original draft, Writing - review and editing; Joo Heon Shin, Software, Supervision, Investigation, Visualization, Methodology; Gabsang Lee, Conceptualization, Data curation, Formal analysis, Supervision, Funding acquisition, Investigation, Methodology, Writing - original draft, Project administration, Writing - review and editing

## Author ORCIDs

Yohan Oh  https://orcid.org/0000-0002-9249-8664
Bin-Kuan Chou  https://orcid.org/0000-0001-6022-4127
Hyesoo Kim  http://orcid.org/0000-0002-4314-1327
Gabsang Lee  https://orcid.org/0000-0002-5052-5927

## Decision letter and Author response

Decision letter https://doi.org/10.7554/eLife.46981.sa1
Author response https://doi.org/10.7554/eLife.46981.sa2

# Additional files

## Supplementary files

• Supplementary file 1. Gene expression levels and GO terms of Five groups during myogenesis, Related to *Figure 2* and *Figure 2—figure supplement 1B*.

• Supplementary file 2. DGEs and GO terms of Three groups and DGEs of Two groups, Related to *Figure 2E* and *Figure 2—figure supplement 1D*.

• Supplementary file 3. Clustering and GO terms of Five Clusters, Related to *Figure 3* and *Figure 3—figure supplement 1*.

• Supplementary file 4. Gene expression levels, DGEs and GO terms of Three groups and DGEs of TWIST1 KO lines, Related to *Figure 6* and *Figure 6—figure supplement 1*.

• Supplementary file 5. Resources table for the primers.

• Transparent reporting form

## Data availability

All the RNA-seq data were deposited to NCBI (GSE129505). The access to the data is available to public.

The following dataset was generated:

| Author(s) | Year | Dataset title | Dataset URL | Database and Identifier |
|---|---|---|---|---|
| Lee G, Shin J, Choi IY | 2019 | Transcriptional landscape of human myogenesis reavels a key role of TWIST1 in maintenance of skeletal muscle progenitors | https://www.ncbi.nlm.nih.gov/geo/query/acc.cgi?acc=GSE129505 | NCBI Gene Expression Omnibus, GSE129505 |

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
