## [Decision Letter]

**Acceptance summary:**

This study characterises distinct stages of myogenic cell production from human pluripotent stem cells from OCT4+ stage to the generation of differentiated MYOGENIN-expressing cells. The authors provide detailed transcriptomics data of the dynamics of lineage progression in this system. In doing so, they identify TWIST1 as a key regulator for maintenance of PAX7+ skeletal muscle progenitors. Therefore, this study provides important information of stage-specific generation of skeletal muscles from human pluripotent cells.

**Decision letter after peer review:**

Thank you for submitting your article "Transcriptional landscape of human myogenesis reveals a key role of TWIST1 in maintenance of skeletal muscle progenitors" for consideration by *eLife*. Your article has been reviewed by three peer reviewers, and the evaluation has been overseen by a Reviewing Editor and Didier Stainier as the Senior Editor. The reviewers have opted to remain anonymous.

The reviewers have discussed the reviews with one another and the Reviewing Editor has drafted this decision to help you prepare a revised submission.

Choi et al. utilize an in vitro differentiation system of human pluripotent stem cells (PSC) to recapitulate key events of skeletal myogenesis. Differentiation protocols for human iPSCs are available in the literature, however a detailed step-wise analysis is lacking. To do so, the authors generated several human eGFP reporter ES cell lines, which allowed for the purification of specific cell populations at different stages of development: these include pluripotent (OCT4), presomitic mesoderm (MSGN1), skeletal muscle progenitors (PAX7), "myoblasts" (MYOG) and multinucleated myotubes. Whole transcriptome analysis using these reporter lines identified selected genes. Validations of some were done (CD271, RUNX1, and TWIST1). Overall the study does not provide major insights into the myogenic process. However, this in vitro PSC differentiation system can be considered to be a useful resource. In this light, there are some major gaps in the analysis that would need to be addressed.

Major points:

General:

A) Protocols for iPSC differentiation still generate heterogenous cell populations, including non-myogenic cells. This point needs to be addressed especially concerning PAX7 which is expressed in neural crest and neuronal cells. Further, there are major gaps in the developmental sequence covered by the study (15-20 days). MSGN1 reporter positive cells are MSGN1+:TBX6+ cells, which represent the paraxial mesoderm before the formation of somites. Therefore, in between the MSGN1+ stage and the PAX7+ stage, the intermediate somite stage is missing. Moreover, although dermomyotome progenitors are PAX7+ and negative for muscle determination gene (MYF5, MYOD) expression – the authors need to consider that at least in the mouse Pax3 and MYF5 expression precede that of PAX7. After a primary myotome is formed, PAX7 then contributes to further myogenesis.

A careful time course analysis is required to assess if this is also the case in human and if Pax3 could provide an intermediate link. Also, authors should consider the expression of other genes (see below) that might bridge the gap. Based on the RNA expression data, showing MYF5 and MYOD expression, it is possible that PAX7 reporter+ cells are a mixed population. A higher resolution analysis could resolve the missing transition states and clarify some of these issues.

Specifically, the authors should:

– report the proportion of PAX7 reporter+ cells expressing MYOD protein.

– characterise cultures between MSGN1+ and PAX7+ states, including Meox1, FoxC 1/2, Paraxis (scleraxis – dermomyotome derivative) and Pax3 expression. While including these stages will make the study comprehensive, the lack of the somite stage may not seriously limit the usefulness of study as a resource. It is important to thoroughly characterise the steps in between and explicitly state the steps missing in the data.

B) For this study to serve the community as a resource, the authors should demonstrate that mining their data will yield insights. In this report, the data on CD271 and Twist1 are presented as examples of findings that could be made using the data. However, deeper characterisation is needed. In addition, organising the data with a user interface would be useful for exploitation of the data by the research community.

C) The study is largely descriptive, focusing on genes previously reported to be involved in myogenesis (PLoS Genet. 2015, 11(8):e1005457; Sci Rep. 2018, 8(1):6555; Dev Cell. 2018, 45(6):712-725). Therefore, despite the potential for discovery, the study does not provide significant conceptual advance for the field as the data largely fits with our current understanding of murine myogenesis. For example, the GO analyses performed in the study to identify functionally-related genes expressed in the distinct developmental stages does not seem to reveal any novel information from the time points chosen.

D) While the study highlights transcription factors expressed differentially in the discrete stages, analysing whether they act in the network to control the different processes associated with each stage will add useful information.

Specific:

1) Figure 4A-F, CD271 (mesenchymal stromal marker) was identified as a cell surface marker to isolate PAX7+ cells suggesting that it is a new marker for muscle progenitor cells in this manuscript. However, CD271(NGFR) was previously identified as a marker for skeletal muscle progenitors in hPSCs. Please cite and discuss the paper published by Hicks et al., 2018; Sakai-Takemura et al., 2018.

Also, the authors clearly show that CD271 is highly expressed in the PAX7+ fraction, but they do not show the expression pattern in the PAX7neg fraction. It is important to further confirm the usefulness of this antibody for the purification of myogenic progenitors. Please compare unsorted, PAX7+, and PAX7neg cell fractions. The authors show that CD271 bright can be expanded without losing myogenic capability. For this finding to be meaningful, the authors should show that other cell fractions (unsorted, CD271low and CD271inter) do lose myogenic capability upon expansion.

2) The authors attempt to provide more mechanistic insight on the role of TWIST1 in the specification, maintenance and differentiation of PAX7+ cells, but the analysis, unfortunately, remains superficial. Confirmation of TWIST1 deletion and subsequent lower levels of PAX7 should be supported by western blot. The authors should comment on the publication on TWIST1 that suggests a different role for this gene (Dev Cell. 2018, 45(6):712-725). This should be discussed.

3) Figure 1C – FACS analysis shows that MSGN1-EGFP+ reached 94% on day4. But the MSGN1+ and MSGN1- cells population are not well distinguished. Please check gating conditions and use hPSC cell line (without EGFP) as a negative control. Also, how was the FACS done for MF20? This is a late differentiation marker. Was the FACS done on single cells and doublets/triplets? Further, MYOG-EGFP+ cells is around 6% in Figure 1D, but stated as 11.81% in the text (paragraph two subsection “Generation of stage-specific genetic reporter hPSC lines to stimulate human embryonic myogenesis in vitro”). Please explain discrepancy.

4) Figure 1D – shows a time-course expression of each marker gene. However, differently from the authors' hypothesis and Figure 1A, PAX7 expression is not higher and does not appear before MYOG expression, indicating that these myocytes (MYOG+, Day30) were not derived from the PAX7+ cells. Also, contrary to these data, Figure 1B shows PAX7 protein highly expressed on Day 20. Please explain these discrepancies and comment on the variability in experiments.

5) Figure 4G-H – authors claim that RUNX1 is a marker for myotube formation, but not the marker for "myoblasts" (MYOG+). However, in Figure 3B cluster 10, both MYOG and RUNX1 are relatively highly expressed in muscle progenitors (PAX7+) and "myoblasts" (MYOG+). However, in Figure S4D, there is no co-localization between PAX7 and RUNX1 antibody staining. Please explain the difference between Figure 4G and Figure S4D. The authors should check the co-localization of MYOG and RUNX1 by immunostaining to clarify their expression in myoblasts and myotubes.

6) Figure 5D – the population of PAX7+ cells is approximately 10% in both of WT and TWIST1-KO cell lines, which is different from 20% of PAX7+ cell shown in Figure 1C and 17.8% in the Results section of Figure 1. Please comment on the variations between experiments and indicate ranges of expression. Perhaps the WT cell line could reach 20% for PAX7 population (Figure 1C), but TWIST1-KO might have decreased levels of PAX7. Considering TWIST1 is a transcription factor essential for mesoderm development, the expression of MSGN1 (day4), TBX6 (day4), PAX7(day20), MYF5 (day20) should be verified.

7) Figure 5E – please show in detail the culture protocol for muscle progenitor maintenance and images of cultured cells and cell number count from 0P to 4P. If the interpretation is correct, in Figure 5E both WT and KO PAX7+ cell population are remarkably decreased from 0P to 1P; KO even more so (from 100% to 23%) compared to WT (from 100% to 37%). However, by 4P, both WT and KO slightly decreased in PAX7+ cells, and KO have approximately 1/2 PAX7+ compared to WT from 2P-4P. These results suggest that the effect of TWIST1-KO is in the period after sorting (0P-1P), but might not be required later. Please also check the immunostaining with MYOD or MYOG in the maintenance 0P-1P to assess the committed status of PAX7+ cells. It remains possible that TWIST1 plays an indirect role in PAX7 regulation; three is a trend to lower proliferation as seen in Figure 5—figure supplement 1D-E.

8) Figure 5—figure supplement 1A – it is hard to see if TWIST1+ staining co-localizes with DAPI. Please show clearer images.

9) Saethre-Chotzen syndrome patients have prematurely fused skull bones along the coronal suture and significant phenotypes in skeletal muscle are not noted. It is therefore not clear how the data provided in this manuscript explains the mechanism of Saethre-Chotzen syndrome. Furthermore, mutations of *FGFR2* or FGFR3 are also reported to be associated with Saethre-Chotzen syndrome. The authors should omit the reference to Saethre-Chotzen syndrome in the Abstract, and speculate in the Discussion, if this is of relevance.

10) A "myoblast" is generally referred to as a proliferating cell. It is likely that MYOG+ cells are no longer proliferating, and it is a differentiation marker, hence these cells should be referred to as differentiated cells, and staining needs to be done with this reporter gene and a proliferation marker (ex. Ki67, BrdU).

---

## [Author Response]

Major points:General:A) Protocols for iPSC differentiation still generate heterogenous cell populations, including non-myogenic cells. […] A higher resolution analysis could resolve the missing transition states and clarify some of these issues.Specifically, the authors should:– report the proportion of PAX7 reporter+ cells expressing MYOD protein.– characterise cultures between MSGN1+ and PAX7+ states, including Meox1, FoxC 1/2, Paraxis (scleraxis – dermomyotome derivative) and Pax3 expression. While including these stages will make the study comprehensive, the lack of the somite stage may not seriously limit the usefulness of study as a resource. It is important to thoroughly characterise the steps in between and explicitly state the steps missing in the data.

We thank the reviewer for constructive comments. To confirm the cellular identity of the PAX7::EGFP+ cells generated by our skeletal muscle differentiation protocol, we performed RT-PCR with primer sets specific for neural and neural crest lineages such as SOX10, PAX6, and antibody staining of SOX10 and AP2 (Figure 1—figure supplement 2A-C). PAX7::EGFP+ cells showed significantly low levels of gene expression, compared to control neural crest cells (as a positive control). In the protein level, we did not detect any SOX10+ cells, or AP2+ cells in the PAX7::EGFP+ cells. With these results, we demonstrate that PAX7::EGFP+ cells derived from our skeletal muscle protocol are mostly skeletal muscle lineage cells.

As the reviewer pointed out, we confirmed the levels of PAX7, MYOD1, and MYOG protein expression in PAX7::EGFP+ cells (Figure 1—figure supplement 2D) via antibody staining. Most of the PAX7::EGFP+ cells are expressing PAX7 protein, but not MYOD1 and MYOG, although we cannot rule out their possible transcriptional heterogeniety (we performed single cell qRT-PCR with PAX7::EGFP+ cells, please see Figure 1—figure supplement 2F). These data were added in the Results section and Figure 1—figure supplement 2.

For the characterization between MSGN1 and PAX7, we performed the gene expression profiles of T(Brachyury), MIXL1, PAX3, MEOX1, FOXC1, FOXC2, PARAXIS, and SCLERAXIS during in vitro myogenesis (Author response image 1). PAX3, MEOX1, FOXC1, and FOXC2 gene started their gene expression at Day 4, and had a peak between Day 6 and Day 8 which imply that intermediate somite stage fills the gap between MSGN1+ stage and PAX7+ stage. Furthermore, PAX3 gene expression was confirmed through PAX3::EGFP reporter line (Author response image 2). PAX3 gene expression through EGFP protein started at Day 4 and had a peak at Day 8, and disappear gene expression from Day 10. And PAX3 protein expression was also validated during in vitro skeletal muscle specification. We include these new data in the Results and Figure 1—figure supplement 1A.

**Author response image 1. respfig1:** mRNA expression pattern of several marker genes during in vitro skeletal muscle specification (n = 2, independent biological repeats).

**Author response image 2. respfig2:** PAX3 gene expression during in vitro skeletal muscle differentiation. (**A**) EGFP expression pattern for PAX3 gene during skeletal muscle specification (n = 2). (**B**) Immunohistochemistry for the PAX3 and TBX6 at Day 4 (bar, 100μm).

B) For this study to serve the community as a resource, the authors should demonstrate that mining their data will yield insights. In this report, the data on CD271 and Twist1 are presented as examples of findings that could be made using the data. However, deeper characterisation is needed. In addition, organising the data with a user interface would be useful for exploitation of the data by the research community.

Thank you for the constructive comments and concerns. We agree with the comment that deeper characterization will be more informative to identify novel insights. As alluded in the above comment, our current study for human myogenesis has possibilities to identify regulators of human myogenic ontogeny. First, we present new transcription analysis data of differential gene expression between PAX7::EGFP+ cells and MYOG::EGFP+ cells (Figure 2—figure supplement 2A-B). Volcano plot indicated significant change in PAX7::EGFP+ cells and MYOG::EGFP+ cells. Furthermore, we classified highly enriched genes specifically in PAX7::EGFP+ cells, MYOG::EGFP+ cells, and Myotubes. Venn diagram showed clear separation between PAX7::EGFP+ cells and Myotubes, while there are overlapped genes in PAX7::EGFP+ cells vs. MYOG::EGFP+ cells, and MYOG::EGFP+ cells vs. Myotubes. Out of 2336 genes upregulated over all samples, 420 genes have specific expression patterns only in PAX7::EGFP+ cells, which were classified into GO terms involved in some signaling pathways including ‘PI3K-Akt signaling pathway’, ‘Wnt signaling’, and ‘Wnt signaling and pluripotency’. These data were confirmed via a phosphorylation antibody blot (R&D, ARY003B), presenting that significantly increased levels of phosphorylation in the CREB and β-catenin in the PAX7::EGFP+ cells over the MYOG::EGFP+ cells (Figure 2—figure supplement 2C). Thanks to the reviewer, we identified ‘Wnt signaling’ pathway is a close association in PAX7::EGFP+ cells, which is consistent with our RNA-seq analysis with TWIST1 KO PAX7::EGFP+ cells (Figure 6D). The detailed experiments for the ‘Wnt signaling’ pathway in early human myogenesis will be pursued in future studies. We include these data in the Results and Figure 2—figure supplement 2A-C.

As suggested in Question B, we organized the data with a user-friendly interface and created a website with an easily searchable interface to act as a companion to this resource paper (www.myogenesis.net) (a screenshot of the website is in Author response image 3). This website will be maintained and updated as new information becomes available. This information was added in the Abstract.

**Author response image 3. respfig3:** A representative screen-shot of inclusive gene expression atlas during human myogenesis.

C) The study is largely descriptive, focusing on genes previously reported to be involved in myogenesis (PLoS Genet. 2015, 11(8):e1005457; Sci Rep. 2018, 8(1):6555; Dev Cell. 2018, 45(6):712-725). Therefore, despite the potential for discovery, the study does not provide significant conceptual advance for the field as the data largely fits with our current understanding of murine myogenesis. For example, the GO analyses performed in the study to identify functionally-related genes expressed in the distinct developmental stages does not seem to reveal any novel information from the time points chosen.

This is a valid point. In this study, our primary goal was realizing comprehensive characterization of the myogenic population derived from human pluripotent stem cells. One of the advantages of our system is time-course analysis of the transcriptional landscape mimicked in vivo human myogenesis, and K-mean clustering grouped genes with significant changes in the time dependent expressions with distinct transcriptional variations during in vitro human skeletal muscle specification. These data defined specialized signature expression profiles for each isolated cell type, constituted of genes with the potential to control cell fates during human skeletal muscle specification. As recommended, to investigate the changes in gene expression associated with the myotube formation, we additionally compared global expression profiles of MYOG::EGFP+ cells and Myotubes (Author response image 4). Ras Pathway, FGF signaling pathway, and EGF receptor signaling pathway were involved in MYOG::EGFP+ cells. Furthermore, we performed differential gene expression analysis between PAX7::EGFP+ cells and MYOG::EGFP+ cells (Figure 2—figure supplement 2A-B). In these data, PI3K-Akt signaling pathway, Wnt signaling, and Wnt signaling and pluripotency were involved in PAX7::EGFP+ cells. With the increased levels of phosphorylation in PAX7::EGFP+ cells via MYOG::EGFP+ cells (Figure 2—figure supplement 2C), we imply that phosphorylation of CREB and β-catenin in PAX7::EGFP+ cells is important to activate CREB during skeletal muscle specification (Stewart, 2011 #127).

**Author response image 4. respfig4:** Pathway analysis of cell type specific gene signatures between MYOG::EGFP+ cells and Myotubes.

D) While the study highlights transcription factors expressed differentially in the discrete stages, analysing whether they act in the network to control the different processes associated with each stage will add useful information.

We appreciate the reviewer’s comment. In an attempt to further analyze the stage-specific transcriptome regulation during human myogenesis, we applied weighted gene co-expression network analysis (WGCNA) with five different cell types (Figure 6—figure supplement 2). WGCNA with stages-specific transcriptomes allowed us to define modules of genes that are continuously coregulated and to study their stage-specific variation during skeletal muscle differentiation. By including all expressed genes with expression variation, we identified a total of 96 modules defined as groups of genes coordinately expressed across 20 samples (5 cell types and 4 different repeated samples) (Figure 6—figure supplement 2). In the overall network, we found that CD271 belongs to Module 22, TWIST1 to Module 43, and RUNX1 to Module 3. As a key strength of WGCNA is the ability to query the network around genes known to be associated with a trait, we examined the genetic associations within the context of their module, as a network with proteins (genes in each module) as nodes and interaction scores as edges. Our data indicated that TWIST1 has direct interactions with 8 well known transcription factors including MYOD1, TCF4, and TP53, suggesting that TWIST1 may play significant roles in the transcriptional network to control the myogenic specification processes (Figure 6—figure supplement 2B). CD271 is a genetic hub for directing 26 genes to over 10 different genetic hubs (Figure 6—figure supplement 2B). RUNX1 showed massive interactions with 32 transcription factors of many different pathways including epigenetic changes and NOTCH signaling related pathway (Figure 6—figure supplement 2B). Taken together, these data demonstrate that TWIST1, CD271 and RUNX1 play significant roles in the transcriptional network to control the myogenic specification processes. These results were added in Discussion and Figure 6—figure supplement 2.

Specific:1) Figure 4A-F, CD271 (mesenchymal stromal marker) was identified as a cell surface marker to isolate PAX7+ cells suggesting that it is a new marker for muscle progenitor cells in this manuscript. However, CD271(NGFR) was previously identified as a marker for skeletal muscle progenitors in hPSCs. Please cite and discuss the paper published by Hicks M, et al. Nat. Cell Biol. 2018; Sakai-Takemura et al. Sci. Reports 2018.Also, the authors clearly show that CD271 is highly expressed in the PAX7+ fraction, but they do not show the expression pattern in the PAX7neg fraction. It is important to further confirm the usefulness of this antibody for the purification of myogenic progenitors. Please compare unsorted, PAX7+, and PAX7neg cell fractions. The authors show that CD271 bright can be expanded without losing myogenic capability. For this finding to be meaningful, the authors should show that other cell fractions (unsorted, CD271low and CD271inter) do lose myogenic capability upon expansion.

We thank the reviewer for the invaluable comments. As mentioned in Question 1, we include two references (Hicks et al., 2018 and Sakai-Takemura et al., 2018) in Discussion.

In the FACS plot, we found three distinctively separate populations in CD271 stained sample (APC only control is in left), and named them as CD271low, CD271inter and CD271bright. Then, we subgated the populations in the PAX7::EGFP channel. As shown in the bottom panels in Figure 4—figure supplement 1A, CD271bright cells included PAX7::EGFP+ cells (77.8% ), compared to the CD271low (2.43%) and CD271inter(24.6% ). With this reason, we only used CD271bright cell populations as a substitute for the PAX7::EGFP+ cells. In the study, we analyzed myogenic capabilities of sorted cells based on CD271 expression levels. As shown Figure 4—figure supplement 1C-E, all three populations showed comparable levels of growth rates, based on the number of cells during cell expansion culture (over two weeks). Importantly, the fusion ability of CD271bright, CD271inter, and CD271low cell populations are quite different, showing that CD271bright cells showed robust fusion capability (comparable to that of PAX7::EGFP+ cells), but not CD271low and CD271inter cells. These data were included in the Results section and Figure 4—figure supplement1.

2) The authors attempt to provide more mechanistic insight on the role of TWIST1 in the specification, maintenance and differentiation of PAX7+ cells, but the analysis, unfortunately, remains superficial. Confirmation of TWIST1 deletion and subsequent lower levels of PAX7 should be supported by western blot. The authors should comment on the publication on TWIST1 that suggests a different role for this gene (Dev Cell. 2018, 45(6):712-725). This should be discussed.

Thank you for pointing this out. As suggested, we confirmed TWIST1 depletion by western blot (Figure 5—figure supplement 1A).

As shown in Author response image 5, PAX7 showed comparable expression level at Day 30 of myogenic specification. After cell sorting, PAX7 expression was decreased in TWIST1 KO PAX7::GFP+ cell clones (#3 and #7), which is a similar pattern to FACS analysis data during expansion of PAX7::GFP+ cells. As mentioned in Question 2, we include the reference (Parajuli et al., 2018) in Discussion. The transcription factors, JUN, ZEB1, ATF-2, MYOD, p53, Pax- 3, STAT3, are known to be bound to transcription factor binding sites in the TWIST1 locus, suggesting that TWIST1 may play a role in myogenic specification processes. This can be one of plausible explanations for our TWIST1 KO experiment results.

**Author response image 5. respfig5:** Western blot of PAX7 in WT and TWIST1 KO lines. (**A**) Western blot of PAX7 in WT and TWIST1 KO lines (clone #3) at Day 30 of myogenic specification. (**B**) Western blot of PAX7 in WT and TWIST1 KO lines (clone #3 and #7) during expansion of PAX7::GFP+ cells after cell sorting at passage 4.

*3) Figure 1C – FACS analysis shows that MSGN1-EGFP+ reached 94% on day4. But the MSGN1+ and MSGN1- cells population are not well distinguished. Please check gating conditions and use hPSC cell line (without EGFP) as a negative control. Also, how was the FACS done for MF20? This is a late differentiation marker. Was the FACS done on single cells and doublets/triplets? Further, MYOG-EGFP+ cells is around 6% in Figure 1D, but stated as 11.81% in the text (paragraph two subsection “Generation of stage-specific genetic reporter hPSC lines to stimulate human embryonic myogenesis* in vitro*”). Please explain discrepancy.*

We apologize for the lack of clarity. We included FACS plots of non-genetically modified parental cell line (H9 cell line) as a negative control (Author response image 6). And our FACS experiment protocol was added in the Materials and methods section.

The data in Figure 1D were measured till Day 30 of myogenic specification (Author response image 6), while the percentages of MYOG::EGFP+ cells in subsection “Generation of stage-specific genetic reporter hPSC lines to simulate human embryonic myogenesis in vitro” were measured at Day 35 (Author response image 6), so there is some temporal gap between the figure panel and the text. We ensured that the dates are included in figure legend and main text. As reviewer pointed out, we added protocol for FACS analysis in the Materials and methods.

“For the MF20 FACS analysis, cells were dissociated with Accutase and MF20 was applied and incubated 20 min at 25°C. Secondary antibody was used for the detecting by BD FACS Calibur (Becton Dickinson), and data were visualized using FlowJo software (Tree Star Inc).”

**Author response image 6. respfig6:** FACS plots of differentiated H9 cells as a negative control to gate MSGN1::EGFP+ cell population. Right panel is a captured image of Figure 1C.

4) Figure 1D – shows a time-course expression of each marker gene. However, differently from the authors' hypothesis and Figure 1A, PAX7 expression is not higher and does not appear before MYOG expression, indicating that these myocytes (MYOG+, Day30) were not derived from the PAX7+ cells. Also, contrary to these data, Figure 1B shows PAX7 protein highly expressed on Day 20. Please explain these discrepancies and comment on the variability in experiments.

Thank you for the constructive comments. We believe that the myogenic specification from human pluripotent stem cells are not synchronized events, as we found a gradual increase of PAX7::EGFP expression during the time course. Also the in vitro condition may not be favourable to the maintenance of PAX7 expression, which forces the PAX7::EGFP+ cells to be differentiated into MYOG::EGFP+ cells. In addition, as shown in Figure 1—figure supplement 2F, single cell transcription analysis data show that some of PAX7::EGFP+ cells already have transcription of MYOG as well as other marker genes, including MYF5 and MYOD1, while most of the PAX7::EGFP+ cells have high levels of PAX7 expression. Therefore, we believe that in vitro culture condition (favorable for differentiation) and the transcriptional heterogeneity of PAX7::EGFP+ cells are responsible for the slightly increased levels of MYOG::EGFP+ cells in early myogenic specification stage (Day 20-22) in the Figure 1D. In other words, our data do not necessarily conclude that MYOG+ cells are not derived from PAX7::EGFP+ cells. We include these data in the Results section and Figure 1—figure supplement 2F, and interpretation in Discussion as below.

“We believe that the myogenic specification from human pluripotent stem cells are not synchronized events, as we see that gradual increase of PAX7::EGFP expression during the time course. Also the in vitro condition may not be favorable to the maintenance of PAX7 expression, which forces the PAX7::EGFP+ cells to be differentiated into MYOG::EGFP+ cells. Therefore, we believe that in vitro culture condition (favorable for differentiation) and the transcriptional heterogeneity of PAX7::EGFP+ cells are responsible for the slightly increased levels of MYOG::EGFP+ cells in early myogenic specification stage (Day 20-22) in the Figure 1D. However, our data do not necessarily conclude that MYOG+ cells are not derived from PAX7::EGFP+ cells.“

5) Figure 4G-H – authors claim that RUNX1 is a marker for myotube formation, but not the marker for "myoblasts" (MYOG+). However, in Figure 3B cluster 10, both MYOG and RUNX1 are relatively highly expressed in muscle progenitors (PAX7+) and "myoblasts" (MYOG+). However, in Figure S4D, there is no co-localization between PAX7 and RUNX1 antibody staining. Please explain the difference between Figure 4G and Figure S4D. The authors should check the co-localization of MYOG and RUNX1 by immunostaining to clarify their expression in myoblasts and myotubes.

This is a valid point. RUNX1 gene is included in the Cluster 10. Cluster 10 is a collection of genes that are not detected in the undifferentiated pluripotent stem cell stage, then start to express during skeletal muscle specification, and show maximum levels of gene expression in the Myotube stage. Indeed, our RNA-seq analysis show that RUNX1 has a gradual expression pattern, detected in PAX7::EGFP+ cells and MYOG::EGFP+ cells, and then highest levels in Myotube group. Furthermore, we tested if protein expression of RUNX1 can be detected in MYOG+ cells, by performing RUNX1 and MYOG antibody staining. As shown in Figure 4—figure supplement 1H, RUNX1 expression was found in multinucleated myotube, and we found that RUNX1 is colocalized MYOG+ cells. These data may imply that there might be a posttranscriptional regulation of RUNX1 during human myogenesis, which results in lack of PAX7 co-localization in myotube stages. The result was added in Figure 4—figure supplement 1H.

6) Figure 5D – the population of PAX7+ cells is approximately 10% in both of WT and TWIST1-KO cell lines, which is different from 20% of PAX7+ cell shown in Figure 1C and 17.8% in the Results section of Figure 1. Please comment on the variations between experiments and indicate ranges of expression. Perhaps the WT cell line could reach 20% for PAX7 population (Figure 1C), but TWIST1-KO might have decreased levels of PAX7. Considering TWIST1 is a transcription factor essential for mesoderm development, the expression of MSGN1 (day4), TBX6 (day4), PAX7(day20), MYF5 (day20) should be verified.

We apologize for the lack of clarity. Usually, we found that PAX7::EGFP+ cells show around 20% of EGFP+ cells at Day 35 of myogenic specification. But we performed the comparison between the wild type (WT) and TWIST1 KO cells in much earlier time points (Day 25 to Day 30) to avoid any possible misinterpretation and/or unexpected effects caused by depleted TWIST1 expression, for example, spontaneous/precocious differentiation to multinucleated myotubes, lack of PAX7 maintenance, or gradual cell death. We presented the data of individual clones of TWIST1 KO lines in Figure 5—figure supplement 1C, showing there is no significant difference in terms of the levels of PAX7::EGFP+ cells.

As reviewer’s recommendation, we performed transcriptional analysis during early development of the TWIST1 KO clones. As shown in Figure 5—figure supplement 2A, MSGN1 and TBX6 gene expression at Day 2 and Day 4 showed comparable pattern to WT and KO cell lines. However, TWIST1 KO clones show aberrant expression patterns in the PAX3, PAX7, MYF5, MYOD1, and MYOG. We included a new sentence in the Results as “there are aberrant expression patterns of PAX3, PAX7, MYF5, and MYOD1 during the myogenic specification, although the levels of PAX7::EGFP+ cells in WT and TWIST1 KO clones (at Day 25 to Day 30 of myogenic specification) are comparable.” We include these data in Figure 5—figure supplement 1C and Figure 5—figure supplement 2A.

7) Figure 5E – please show in detail the culture protocol for muscle progenitor maintenance and images of cultured cells and cell number count from 0P to 4P. If the interpretation is correct, in Figure 5E both WT and KO PAX7+ cell population are remarkably decreased from 0P to 1P; KO even more so (from 100% to 23%) compared to WT (from 100% to 37%). However, by 4P, both WT and KO slightly decreased in PAX7+ cells, and KO have approximately 1/2 PAX7+ compared to WT from 2P-4P. These results suggest that the effect of TWIST1-KO is in the period after sorting (0P-1P), but might not be required later. Please also check the immunostaining with MYOD or MYOG in the maintenance 0P-1P to assess the committed status of PAX7+ cells. It remains possible that TWIST1 plays an indirect role in PAX7 regulation; three is a trend to lower proliferation as seen in Figure 5—figure supplement D-E.

We appreciate comments and concerns. Maintaining skeletal muscle stem cells provide a unique opportunity to understand mechanism of skeletal muscle generation and can be used for further studies, such as in vitro manipulation and disease modeling. In this study, we developed a new culture condition for maintaining putative human PAX7::EGFP+ skeletal muscle stem cells. Although often in vitro culture conditions are permeable to be differentiated into multinucleated myotubes, we found a condition to maintain the around 50% of PAX7::EGFP+ cells with great myogenic capabilities until passage 5 (over a month). In this condition, we found that the EGFP+ levels of TWIST1 KO PAX7::EGFP+ cells could not be maintained. To quantify this trend, we measured percentages of PAX7::EGFP+ cells during the passaging in WT and the KO clones. We present the data of each clone individually (Figure 5—figure supplement 2B) (the data in Figure 5E are combined one of two different KO clones). As shown in Figure 5—figure supplement 2B, both KO clones (clone #1 and #3) gradually lost the GFP levels of PAX7::EGFP+ cells during the passages. In addition, we looked into the cellular aspects. As shown in Figure 5—figure supplement 2C-E, we could not find any significantly difference between the WT and the KO clones in terms of cell morphology, proliferation rates and the proportions of PAX7+ and MYOG+ cells (Figure 5—figure supplement 2C-E). These results were added in the Results and Figure 5—figure supplement 2B-G. As reviewer recommended, we added protocol for cell culture after sorting in the Materials and methods as below.

*“*For the maintinaing EGFP+/- cells after cell sorting, EGFP+/- cells were plated in 1% Geltrexcoated dish with N2 media containing 5% FBS, 10ng/ml FGF-2 and 100ng/ml FGF-8. Media were changed at every other day. For passaging, cells were dissociated with Accutase for 20 min, washed once with N2 media.”

8) Figure 5—figure supplement 1A – it is hard to see if TWIST1+ staining co-localizes with DAPI. Please show clearer images.

Thank you for the pointing this out. As reviewer suggested, we changed the images of TWIST1+ cells (Figure 5—figure supplement 1A), showing that nuclear staining of TWIST1 is obvious in wildtype (WT), but not in knock-out (KO) clones. And these data were included in Figure 5—figure supplement 1A.

9) Saethre-Chotzen syndrome patients have prematurely fused skull bones along the coronal suture and significant phenotypes in skeletal muscle are not noted. It is therefore not clear how the data provided in this manuscript explains the mechanism of Saethre-Chotzen syndrome. Furthermore, mutations of FGFR2 or FGFR3 are also reported to be associated with Saethre-Chotzen syndrome. The authors should omit the reference to Saethre-Chotzen syndrome in the Abstract, and speculate in the Discussion, if this is of relevance.

As recommended, we omit the Saethre-Chotzen syndrome in the Abstract and moved to Discussion part.

10) A "myoblast" is generally referred to as a proliferating cell. It is likely that MYOG+ cells are no longer proliferating, and it is a differentiation marker, hence these cells should be referred to as differentiated cells, and staining needs to be done with this reporter gene and a proliferation marker (ex. Ki67, BrdU).

Thank you for the reviewer’s concern. We checked the proliferation rates by performing Ki67, PAX7, and MYOG antibody staining. As shown in Figure 1—figure supplement 2E, the percentages of Ki67+ cells in PAX7+ and MYOG+ cells are ~35% and ~5% , respectively. As reviewer recommended, we changed the terminology from myoblast to ‘MYOG::EGFP+ cell’.